# Ergodic Performance Analysis of Double Intelligent Reflecting Surfaces-Aided NOMA–UAV Systems with Hardware Impairment

**Minh-Sang Van Nguyen [1]** ![ID], **Dinh-Thuan Do [2]**, **Van-Duc Phan [3,*]** ![ID], **Wali Ullah Khan [4]**,
**Agbotiname Lucky Imoize [5,6]** and **Mostafa M. Fouda [7]** ![ID]

1   Faculty of Electronics Technology, Industrial University of Ho Chi Minh City (IUH),
    Ho Chi Minh City 70000, Vietnam; nguyenvanminhsang@iuh.edu.vn
2   Department of Computer Science and Information Engineering, College of Information and Electrical
    Engineering, Asia University, Taichung 41354, Taiwan; dodinhthuan@asia.edu.tw
3   Faculty of Automotive Engineering, School of Engineering and Technology, Van Lang University,
    Ho Chi Minh City 70000, Vietnam
4   Interdisciplinary Center for Security, Reliability and Trust (SnT), University of Luxembourg,
    1855 Luxembourg, Luxembourg; waliullah.khan@uni.lu
5   Department of Electrical and Electronics Engineering, Faculty of Engineering, University of Lagos,
    Akoka, Lagos 100213, Nigeria; aimoize@unilag.edu.ng
6   Department of Electrical Engineering and Information Technology, Institute of Digital Communication,
    Ruhr University, 44801 Bochum, Germany
7   Department of Electrical and Computer Engineering, College of Science and Engineering,
    Idaho State University, Pocatello, ID 83209, USA; mfouda@ieee.org
*   Correspondence: duc.pv@vlu.edu.vn

**Abstract:** In this work, we design an intelligent reflecting surface (IRS)-assisted Internet of Things (IoT) by enabling non-orthogonal multiple access (NOMA) and unmanned aerial vehicles (UAV) approaches. We pay attention to studying the achievable rates for the ground users. A practical system model takes into account the presence of hardware impairment when Rayleigh and Rician channels are adopted for the IRS–NOMA–UAV system. Our main findings are presented to showcase the exact expressions for achievable rates, and then we derive their simple approximations for a more insightful performance evaluation. The validity of these approximations is demonstrated using extensive Monte Carlo simulations. We confirm the achievable rate improvement decided by main parameters such as the average signal to noise ratio at source, the position of IRS with respect to the source and destination and the number of IRS elements. As a suggestion for the deployment of a low-cost IoT system, the double-IRS model is a reliable approach to realizing the system as long as the hardware impairment level is controlled. The results show that the proposed scheme can greatly improve achievable rates, obtain optimal performance at one of two devices and exhibit a small performance gap compared with the other benchmark scheme.

**Keywords:** intelligent reflecting surface; non-orthogonal multiple access; achievable rate; hardware impairment

## 1. Introduction

In recent years, as a promising transmission method, the new generation wireless systems can rely on IRS and UAV to enhance their spectral and energy efficiency since IRS is examined as a cost-effective deployment approach [1–3]. Specifically, by varying the amplitude and phase for the incident signal, an IRS leverages its low-cost reconfigurable passive elements to reflect signals to distant users effectively [4,5]. The system relying on IRS can enhance the link quality and enlarge the coverage significantly when IRS is able to adjust amplitude-reflection coefficients and phase-shift variables appropriately [6–8].

The authors in [7] studied secure IRS–UAV systems by integrating IRS with UAV in wireless networks, which are susceptible to eavesdropping related to air-ground line-of-sight channels. To facilitate security, IRS can be leveraged due to its capability of reconfiguring the propagation environment, and thus IRS helps to reduce the impacts of eavesdroppers. The secure transmission of an IRS-assisted UAV network is analyzed under the impact of an eavesdropper. The transmit beamforming, the trajectory of UAV and the phase shift of IRS can be optimized to further obtain maximal rates [7]. Besides, IRS has two prominent features—passive reflection with low power consumption and the operation of full-duplex (FD) mode without self-interference—and these benefits are crucial in comparison with conventional approaches—for example, relaying networks [9].

To further increase energy and spectrum efficiency, the power-domain NOMA technology has been studied as a potential technique for IoT applications [10,11]. The NOMA architecture can be described as follows. In the transmit side of a NOMA system, different amounts of power at the transmitter or the base station (BS) are assigned to multiple users when superimposing signal processing, while those users share the same orthogonal resources such as frequency, time and spreading code. At the receiver side associated with a downlink, demultiplexing the transmitted signals is conducted by employing successive interference cancellation (SIC) [12]. To decode the needed signals, the user treats all the weaker user signals as interference; then, its own signal can be decoded in the last step [13–15]. In contrast to the downlink, the common receiver or the BS achieves a signal in an uplink, allowing multiple users to consume the common communication. This means that all users send a superposed signal comprising the signals of these users toward the BS. For signal detection, the BS needs the assistance of SIC to decode the signals of the transmitting users [12]. In this way, a higher number of users can be served in the context of NOMA, which exhibits a significant improvement compared with the conventional orthogonal multiple access (OMA)-aided transmission approach [11].

The combination of IRS and NOMA could be a prominent technique to leverage energy-efficient and low-cost deployment [16–25]. The authors in [16] designed an IRS–NOMA system by considering both continuous phase shifting and discrete phase shifting corresponding to the ideal IRS and the non-ideal IRS circumstance. To demonstrate performance, closed-form expressions are derived to examine the average required transmit power, the outage probability and the diversity order by benefiting from the Laguerre series and the isotropic random vector. In demonstrated analytical results, the BS antenna number and the IRS element number are consistently affected by varying the transmit power. In [17], IRS-aided simultaneous wireless information and power transfer (SWIPT) NOMA networks are studied. In particular, a problem of minimizing the transmit power at the BS can be solved by jointly optimizing the BS transmit beamforming vector, power splitting (PS) ratio, SIC decoding order and IRS phase shift. This optimization is guaranteed to satisfy the energy harvested threshold of each user and the quality-of-service (QoS) constraint. To improve the performance of multiple user equipments (UEs), the BS utilizes the UAV-mounted IRS to flexibly serve ground users [18]. The direct non-line-of-sight links between the BS and UEs are required to power the IRS relaying links. The main performance metric is presented, i.e., the outage probability. The model of IRS–NOMA in [19] allows this cell-edge user to be paired with a cell-center user to form the NOMA scheme. In this way, the coverage can be improved at the cell-edge user. The phase shift plays an important role in the design of an IRS-aided NOMA system, i.e., the impacts of random phase shifting and coherent phase shifting are further investigated [20]. In [21], IRS-assisted NOMA benefits from enabling BS's beamforming vectors. To minimize transmission power, the beamforming vectors and the IRS phase shift matrix could be optimized. One can place the IRS at preferred locations, and line-of-sight (LoS) can be adopted for the links between the transmitters and the IRS before reflecting to receivers [23–25]. The links can be enhanced if the system enables LoS and optimized passive beamforming vectors to implement a NOMA-assisted IRS. In [23], by considering ideal and non-ideal IRSs, the system performance was evalu-

ated for the link from source to destination via IRS. In [24], a tremendous performance improvement could be confirmed in the approach of an IRS-assisted NOMA system. This means that IRS-assisted NOMA exhibits sufficient benefits to be incorporated into the existing IoT systems.

### 1.1. Related Works

In similar work, the authors of [26] presented a hybrid aerial FD relaying protocol consisting of a IRS mounted UAV system. To help the information transfer between the base station and multiple users, the UAV acts as a relay operating in the decode and forward mode. The study also presented the closed-form formulas of achievable throughput, outage probability and ergodic capacity. The authors in [27] studied IRS-assisted multi-user multiple-input single-output (MISO) wireless systems by considering the ergodic capacity in both uplink and downlink scenarios. They examined the realistic case of statistical instantaneous channel state information (CSI). Further, analytical expressions of the ergodic sum capacity were derived as the main contribution. In [28], a two-IRS system was studied by employing the centralized and the distributed modes corresponding to the reflecting elements being mounted at a single IRS, and multiple IRSs were designed with the same number of reflecting elements. To examine the benefits of the two-IRS model, the closed-form approximation expressions of the ergodic capacity were derived along with their tight upper and lower bounds to provide more necessary insights. Although the transmit power and the Rician-$K$ factor have the main influences on the system performance, selected modes of the centralized IRS result in a better ergodic capacity as compared with the distributed IRS mode. It is worth noting that the location of the IRS has strong impacts on ergodic performance. Therefore, a multiple IRS design for IoT is necessary.

### 1.2. Motivations and Our Contributions

In contrast to [27,28], Nakagami-$m$ fading channels are deployed for an IRS-aided system in a recent work [29]. The performance is decided by two phase configuration designs including random and coherent phase shifting. As the main performance metrics, the authors in [29] derived formulas of the outage behavior and the bit error rate when binary modulation schemes are applied. The closed-form approximations for the ergodic capacity are extra performance evaluations. However, we first need to answer how many IRSs are sufficient to obtain the expected achievable rate. Secondly, Rayeligh and Rician channels could be applied for several practical situations of UAV. Therefore, this study prefers to examine the achievable rate in practical scenarios where hardware impairment is a key factor in degraded performance. We can summarize our contributions as follows.

- We consider an IRS–NOMA–UAV system without direct links, which consists of a source and several IRSs. We focus on the performance analysis of a group of two users and further determine the impact of hardware impairment.
- We derive closed-form expressions for the achievable rates for two NOMA users under the channel models of Rayleigh and Rician. Compared with recent work [30], our result could be combined with their result to provide complete ergodic performance analysis in a more practical circumstance.
- We employ Monte Carlo simulations to validate the analytical outage probabilities. The achievable rate of each user mainly depends on power allocation factors rather than other main parameters such as the number of IRSs, the number of meta-surface elements and the IRS reflecting coefficient.

The remainder of this article is organized as follows. In Section 2, a double IRS–NOMA–UAV system model is described, and the respective received signal with hardware impairment is formulated. We provide the derivation of closed-form ergodic achievable rates for a group of IoT users in Section 3. We aim to verify the results of computations in Section 4. Finally, we summarize concluding remarks and future research in Section 5.

## 2. System Model

We aim to design a IoT network by enabling IRS, NOMA and UAV techniques, where a single antenna source ($S$) powers the signal transmission by IRS to serve many users at destinations. The IRSs are mounted in UAVs for better transmission from the base station to ground users. These ground users are divided into many groups, and a considered group contains two users ($D_i; i = 1, 2$). The system can maximize the benefits of IRS if multiple $I$ IRSs are deployed, as shown in Figure 1. It is assumed that there is no direct path between the source $S$ and ground users (destinations). To reduce the cost of the design, we refer to the first scenario where two IRSs, i.e., $I_1$ and $I_2$, possess reflecting elements $N_1$ and $N_2$. Those IRSs are placed at specific locations (i.e., buildings) to reflect passive beamforming signals towards the destinations (IoT devices). We need to answer how many IRSs are required to achieve the best performance at destinations. Therefore, we move our attention to the second scenario (the benchmark) by assigning three IRSs, i.e., $I_1$, $I_2$ and $I_3$, which are installed with reflecting elements $N_1$, $N_2$ and $N_3$, respectively (it is reasonable to design double-IRS due to cost efficiency in deployment. Although multi-IRS was developed in [31], the numerical result demonstrated a small gap between the double-IRS and three-IRSs case. Therefore, in this study, we emphasize the performance for two scenarios of IRSs).

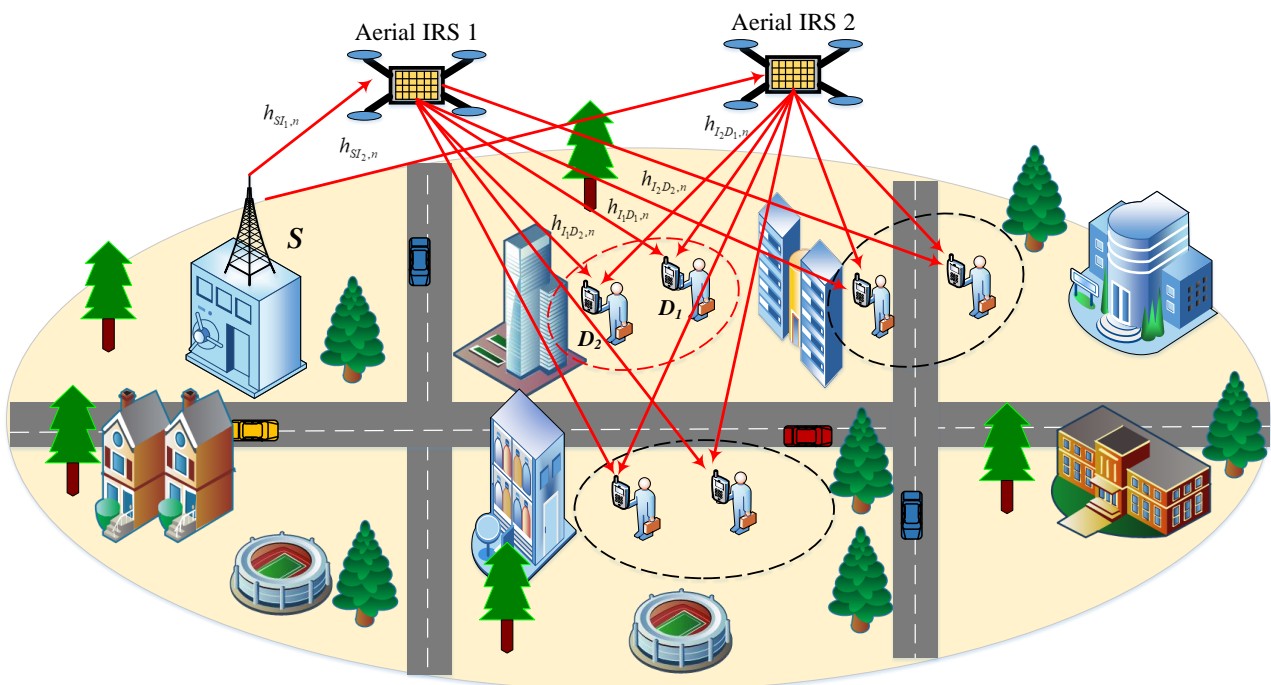

**Figure 1.** The IRS–NOMA–UAV system model with source, two IRSs and ground users.

We denote the (scalar) channels from the $S$ to the IRSs, and from the IRSs to $D_i$, respectively, as $h_{SI_u,n}$ and $h_{I_uD_i,n}$. The hardware impairment at the $S$ is denoted as $\tau_S \sim CN(0, Y_S^2 P)$, where $Y_S$ represents the proportionality coefficients, which describe the severity of the distortion noises at $S$. The hardware impairments at the $D_i$ is $\tau_{D_{ui}} \sim CN\left(0, Y_{D_i}^2 P \left| \sum_{u=1}^{I} \sum_{n=1}^{N_u} h_{SI_u,n} h_{I_uD_i,n} \eta_{un} e^{j\theta_{un}^{(D_i)}} \right|^2 \right)$ [32], where $Y_{D_i}$ represents the proportionality coefficients. Here, $Y_{D_i}$ is associated with the distortion noises at $D_i$. Then, the overall received signal at the user $D_i$ with multiple $I$ IRSs is given as [30,33]

$$y_{D_i} = \sum_{u=1}^{I} \sum_{n=1}^{N_u} h_{SI_u,n} h_{I_uD_i,n} \eta_{un} e^{j\theta_{un}^{(D_i)}} \left( \sqrt{P\chi_1} x_1 + \sqrt{P\chi_2} x_2 + \tau_S \right) + \tau_{D_{ui}} + n_{D_i}, \qquad (1)$$

where $P$ is the total transmit power at the source, $x_i$ is the transmitted signal by source, $\chi_i$ denotes the power allocation factor for message $x_i$ with $\chi_1 + \chi_2 = 1$, $\chi_1 > \chi_2$ [34], $\theta_{un}^{(D_i)}$ is the adjustable phase applied by the $n$-th reflecting element of the $S - I - D_i$, $\eta_{un}$ is the reflection coefficients of $I$ with $\eta_{un} \in (0,1]$, and $n_{D_i}$ stands for the circularly symmetric additive White Gaussian noise (AWGN) with zero mean and variance $\sigma_n^2$, i.e., $n_{D_i} \sim \mathcal{CN}\left(0, \sigma_n^2\right)$.

*2.1. The First Scenario*

The system has two hops from source to IRS and from IRS to destinations. These links are demonstrated in the system model, i.e., $h_{SI_1,n}$, $h_{I_1D_1,n}$, $h_{SI_2,n}$ and $h_{I_2D_1,n}$, along with their channel gains, denoted as $h_{SI_1,n} = d_{SI_1}^{-\alpha/2}\bar{h}_{SI_1,n}e^{-j\phi_{1n}}$, $h_{I_1D_1,n} = d_{I_1D_1}^{-\alpha/2}\bar{h}_{I_1D_1,n}e^{-j\delta_{1n}^{(D_1)}}$, $h_{SI_2,n} = d_{SI_2}^{-\alpha/2}\bar{h}_{SI_2,n}e^{-j\phi_{2n}}$ and $h_{I_2D_1,n} = d_{I_2D_1}^{-\alpha/2}\bar{h}_{I_2D_1,n}e^{-j\delta_{2n}^{(D_1)}}$ [35]; the links $S - I_1$, $I_1 - D_1$, $S - I_2$ and $I_2 - D_1$ are associated with distances $d_{SI_1}$, $d_{I_1D_1}$, $d_{SI_2}$ and $d_{I_2D_1}$, respectively. Here, the path-loss coefficient is $\alpha$, while $\phi_{1n}$ and $\phi_{2n}$ and $\delta_{1n}^{(D_1)}$ and $\delta_{2n}^{(D_1)}$ are the phases of the channel gains. $\bar{h}_{SI_1,n}$, $\bar{h}_{I_1D_1,n}$, $\bar{h}_{SI_2,n}$ and $\bar{h}_{I_2D_1,n}$ are the amplitudes of the channel gains, and they are adopted with a Rayleigh distribution. In the scope of our paper, it is reasonable to assume that the IRS achieves the full knowledge of the channels $h_{SI_1,n}$, $h_{I_1D_1,n}$, $h_{SI_2,n}$ and $h_{I_2D_1,n}$.

The overall received signal at user $D_1$ is given as [36]

$$y_{D_1}^{(1)} = \left( \sum_{n=1}^{N_1} h_{SI_1,n}h_{I_1D_1,n}\eta_{1n}e^{j\theta_{1n}^{(D_1)}} + \sum_{n=1}^{N_2} h_{SI_2,n}h_{I_2D_1,n}\eta_{2n}e^{j\theta_{2n}^{(D_1)}} \right)\left( \sqrt{P\chi_1}x_1 + \sqrt{P\chi_2}x_2 + \tau_S \right) + \tau_{D_{21}} + n_{D_1}, \quad (2)$$

where $\theta_{1n}^{(D_1)}$ is the adjustable phase applied by the $n$th reflecting element of $S - I_1 - D_1$, $\theta_{2n}^{(D_1)}$ stands for the adjustable phase applied by the $n$th reflecting element of $S - I_2 - D_1$, $\eta_{1n}$ is the reflection coefficient of $I_1$ with $\eta_{1n} \in (0,1]$, and $\eta_{2n}$ represents the reflection coefficients of $I_2$ with $\eta_{2n} \in (0,1]$.

The hardware impairment at the $D_1$ is denoted as [32]

$$\tau_{D_{21}} \sim CN\left( 0, Y_{D_1}^2 P\left| \sum_{n=1}^{N_1} h_{SI_1,n}h_{I_1D_1,n}\eta_{1n}e^{j\theta_{1n}^{(D_1)}} + \sum_{n=1}^{N_2} h_{SI_2,n}h_{I_2D_1,n}\eta_{2n}e^{j\theta_{2n}^{(D_1)}} \right|^2 \right). \quad (3)$$

The received signal to noise plus distortion ratio (SNDR) at the destination $D_1$ is defined as

$$\gamma_{D_1}^{(1)} = \frac{\gamma\chi_1 \left| A_1^{(1)}e^{j\left(\theta_{1n}^{(D_1)}-\phi_{1n}-\delta_{1n}^{(D_1)}\right)} + A_2^{(1)}e^{j\left(\theta_{2n}^{(D_1)}-\phi_{2n}-\delta_{2n}^{(D_1)}\right)} \right|^2}{\gamma\chi_2 \left| A_1^{(1)}e^{j\left(\theta_{1n}^{(D_1)}-\phi_{1n}-\delta_{1n}^{(D_1)}\right)} + A_2^{(1)}e^{j\left(\theta_{2n}^{(D_1)}-\phi_{2n}-\delta_{2n}^{(D_1)}\right)} \right|^2 + \left(Y_S^2 + Y_{D_1}^2\right)\gamma \left| A_1^{(1)}e^{j\left(\theta_{1n}^{(D_1)}-\phi_{1n}-\delta_{1n}^{(D_1)}\right)} + A_2^{(1)}e^{j\left(\theta_{2n}^{(D_1)}-\phi_{2n}-\delta_{2n}^{(D_1)}\right)} \right|^2 + 1}, \quad (4)$$

where $\gamma = \frac{P}{\sigma_n^2}$ is the average signal-to-noise ratio (SNR) at the source, $A_1^{(1)} = d_{SI_1}^{-\alpha/2}d_{I_1D_1}^{-\alpha/2}$ $\sum_{n=1}^{N_1}\bar{h}_{SI_1,n}\bar{h}_{I_1D_1,n}\eta_{1n}$, $A_2^{(1)} = d_{SI_2}^{-\alpha/2}d_{I_2D_1}^{-\alpha/2}\sum_{n=1}^{N_2}\bar{h}_{SI_2,n}\bar{h}_{I_2D_1,n}\eta_{2n}$, $\eta_{1n} = \eta, \forall_{1n}$ and $\eta_{2n} = \eta, \forall_{2n}$.

It is noted that from (4) that the highest value of $\gamma_{D_1}^{(1)}$ can be obtained by disregarding the channel phases. In this case, the phases can be modified as $\theta_{1n}^{(D_1)} = \phi_{1n} + \delta_{1n}^{(D_1)}$ for

$1n = 1, \ldots, N_1$ and $\theta_{2n}^{(D_1)} = \phi_{2n} + \delta_{2n}^{(D_1)}$ for $2n = 1, \ldots, N_2$ [35], while the maximal $\gamma_{D_1}^{(1)}$ can be written as

$$\gamma_{D_1}^{(1)} = \frac{\gamma\chi_1 \left| A_1^{(1)} + A_2^{(1)} \right|^2}{\gamma\chi_2 \left| A_1^{(1)} + A_2^{(1)} \right|^2 + \left( Y_S^2 + Y_{D_1}^2 \right)\gamma \left| A_1^{(1)} + A_2^{(1)} \right|^2 + 1}. \tag{5}$$

The achievable rate for the $D_1$ is given as

$$R_{D_1}^{(1)} = \log_2\left( 1 + \gamma_{D_1}^{(1)} \right). \tag{6}$$

The overall received signal at the user $D_2$ is given as [36]

$$y_{D_2}^{(1)} = \left( \sum_{n=1}^{N_1} h_{SI_1,n} h_{I_1 D_2,n} \eta_{1n} e^{j\theta_{1n}^{(D_2)}} + \sum_{n=1}^{N_2} h_{SI_2,n} h_{I_2 D_2,n} \eta_{2n} e^{j\theta_{2n}^{(D_2)}} \right) \left( \sqrt{P\chi_1} x_1 + \sqrt{P\chi_2} x_2 + \tau_S \right) + \tau_{D_{22}} + n_{D_2}, \tag{7}$$

where $\theta_{1n}^{(D_2)}$ stands for the adjustable phase applied by the $n$th reflecting element of the $S - I_1 - D_2$, and $\theta_{2n}^{(D_2)}$ is the adjustable phase applied by the $n$th reflecting element of the $S - I_2 - D_2$. In addition, $h_{I_1 D_2,n}$ and $h_{I_2 D_2,n}$ are the channel gains with $h_{I_1 D_2,n} = d_{I_1 D_2}^{-\alpha/2} \bar{h}_{I_1 D_2,n} e^{-j\delta_{1n}^{(D_2)}}$ and $h_{I_2 D_2,n} = d_{I_2 D_2}^{-\alpha/2} \bar{h}_{I_2 D_2,n} e^{-j\delta_{2n}^{(D_2)}}$ [35], where $d_{I_1 D_2}$ and $d_{I_2 D_2}$ are the distances for the $I_1 - D_2$ link and $I_2 - D_2$ link, respectively, and $\delta_{1n}^{(D_2)}$ and $\delta_{2n}^{(D_2)}$ are the phases of the channel gains. $\bar{h}_{I_1 D_2,n}$ and $\bar{h}_{I_2 D_2,n}$ following a Rayleigh distribution are the amplitudes of the channel gains. Moreover, we assume that the IRS has the full knowledge of the channel phases of $h_{I_1 D_2,n}$ and $h_{I_2 D_2,n}$. The hardware impairments at the $D_2$ is denoted as [35]

$$\tau_{D_{22}} \sim CN\left( 0, Y_{D_2}^2 P \left| \sum_{n=1}^{N_1} h_{SI_1,n} h_{I_1 D_2,n} \eta_{1n} e^{j\theta_{1n}^{(D_2)}} + \sum_{n}^{N_2} h_{SI_2,n} h_{I_2 D_2,n} \eta_{2n} e^{j\theta_{2n}^{(D_2)}} \right|^2 \right). \tag{8}$$

The resulting SNDR at the second user $D_2$ to decode $x_1$ can be formulated as

$$\gamma_{x_1,D_2}^{(1)} = \frac{\gamma\chi_1 \left| B_1^{(1)} e^{j\left( \theta_{1n}^{(D_2)} - \phi_{1n} - \delta_{1n}^{(D_2)} \right)} + B_2^{(1)} e^{j\left( \theta_{2n}^{(D_2)} - \phi_{2n} - \delta_{2n}^{(D_2)} \right)} \right|^2}{\gamma\chi_2 \left| B_1^{(1)} e^{j\left( \theta_{1n}^{(D_2)} - \phi_{1n} - \delta_{1n}^{(D_2)} \right)} + B_2^{(1)} e^{j\left( \theta_{2n}^{(D_2)} - \phi_{2n} - \delta_{2n}^{(D_2)} \right)} \right|^2 + \left( Y_S^2 + Y_{D_2}^2 \right)\gamma \left| B_1^{(1)} e^{j\left( \theta_{1n}^{(D_2)} - \phi_{1n} - \delta_{1n}^{(D_2)} \right)} + B_2^{(1)} e^{j\left( \theta_{2n}^{(D_2)} - \phi_{2n} - \delta_{2n}^{(D_2)} \right)} \right|^2 + 1}, \tag{9}$$

where $B_1^{(1)} = d_{SI_1}^{-\alpha/2} d_{I_1 D_2}^{-\alpha/2} \eta \sum_{n=1}^{N_1} \bar{h}_{SI_1,n} \bar{h}_{I_1 D_2,n}$, $B_2^{(1)} = d_{SI_2}^{-\alpha/2} d_{I_2 D_2}^{-\alpha/2} \eta \sum_{n=1}^{N_2} \bar{h}_{SI_2,n} \bar{h}_{I_2 D_2,n}$.

In (9), to obtain the maximum value of $\gamma_{x_1,D_2}^{(1)}$, we need to eliminate the channel phases. Similar to the method in [35], by setting the phases $\theta_{1n}^{(D_2)} = \phi_{1n} + \delta_{1n}^{(D_2)}$ and $\theta_{2n}^{(D_2)} = \phi_{2n} + \delta_{2n}^{(D_2)}$, the maximal $\gamma_{x_1,D_2}^{(1)}$ can be written as

$$\gamma_{x_1,D_2}^{(1)} = \frac{\gamma\chi_1 \left| B_1^{(1)} + B_2^{(1)} \right|^2}{\gamma\chi_2 \left| B_1^{(1)} + B_2^{(1)} \right|^2 + \left( Y_S^2 + Y_{D_2}^2 \right)\gamma \left| B_1^{(1)} + B_2^{(1)} \right|^2 + 1}. \tag{10}$$

After SIC, the resulting SNDR at the user $D_2$ to decode $x_2$ can be formulated as

$$\gamma_{x_2,D_2}^{(1)} = \frac{\gamma\chi_2 \left| B_1^{(1)} + B_2^{(1)} \right|^2}{\left( Y_S^2 + Y_{D_2}^2 \right)\gamma \left| B_1^{(1)} + B_2^{(1)} \right|^2 + 1}. \tag{11}$$

The achievable rate for the $D_2$ is given as

$$R_2^{(1)} = \log_2\left(1 + \min\left\{\gamma_{x_1,D_2}^{(1)}, \gamma_{x_2,D_2}^{(1)}\right\}\right). \tag{12}$$

### 2.2. The Second Scenario (the Benchmark)

The overall received signal at the user $D_1$ is given as [33,36]

$$y_{D_1}^{(2)} = \left(\sum_{n=1}^{N_1} h_{SI_1,n} h_{I_1D_1,n} \eta_{1n} e^{j\theta_{1n}^{(D_1)}} + \sum_{n=1}^{N_2} h_{SI_2,n} h_{I_2D_1,n} \eta_{2n} e^{j\theta_{2n}^{(D_1)}} + \sum_{n=1}^{N_3} h_{SI_3,n} h_{I_3D_1,n} \eta_{3n} e^{j\theta_{3n}^{(D_1)}}\right)\left(\sqrt{P\chi_1}x_1 + \sqrt{P\chi_2}x_2 + \tau_S\right) \tag{13}$$
$$+ \tau_{D_{31}} + n_{D_1},$$

where $\tau_{D_{31}} \sim CN\left(0, Y_{D_1}^2 P\left|\sum_{n=1}^{N_1} h_{SI_1,n} h_{I_1D_1,n} \eta_{1n} e^{j\theta_{1n}^{(D_1)}} + \sum_{n=1}^{N_2} h_{SI_2,n} h_{I_2D_1,n} \eta_{2n} e^{j\theta_{2n}^{(D_1)}}\right.\right.$

$\left.\left.+ \sum_{n=1}^{N_3} h_{SI_3,n} h_{I_3D_1,n} \eta_{3n} e^{j\theta_{3n}^{(D_1)}}\right|^2\right)$. In addition, $h_{SI_3,n}$ and $h_{I_3D_1,n}$ are the channel gains with

$h_{SI_3,n} = d_{SI_3}^{-\alpha/2}\bar{h}_{SI_3,n}e^{-j\phi_{3n}}$, $h_{I_3D_1,n} = d_{I_3D_1}^{-\alpha/2}\bar{h}_{I_3D_1,n}e^{-j\delta_{3n}^{(D_1)}}$ [35], where $d_{SI_3}$ and $d_{I_3D_1}$ are the distances for the $S - I_3$ link, $I_3 - D_1$ link, respectively, and $\phi_{3n}$ and $\delta_{3n}^{(D_1)}$ are the phases of the channel gains. $\bar{h}_{SI_3,n}$ and $\bar{h}_{I_3D_1,n}$ following a Rayleigh distribution are the amplitudes of the channel gains. Moreover, we assume that the IRS has full knowledge of the channel phases of $h_{SI_3,n}$, $h_{I_3D_1,n}$.

Similar to (4), the received SNDR at the destination $D_1$ is defined as

$$\gamma_{D_1}^{(2)} = \frac{\gamma\chi_1\left|A_1^{(1)} + A_2^{(1)} + A_3^{(1)}e^{j\left(\theta_{3n}^{(D_1)} - \phi_{3n} - \delta_{3n}^{(D_1)}\right)}\right|^2}{\gamma\chi_2\left|A_1^{(1)} + A_2^{(1)} + A_3^{(1)}e^{j\left(\theta_{3n}^{(D_1)} - \phi_{3n} - \delta_{3n}^{(D_1)}\right)}\right|^2 + \left(Y_S^2 + Y_{D_1}^2\right)\gamma\left|A_1^{(1)} + A_2^{(1)} + A_3^{(1)}e^{j\left(\theta_{3n}^{(D_1)} - \phi_{3n} - \delta_{3n}^{(D_1)}\right)}\right|^2 + 1}, \tag{14}$$

where $A_3^{(1)} = d_{SI_3}^{-\alpha/2} d_{I_3D_1}^{-\alpha/2} \sum_{n=1}^{N_3} \bar{h}_{SI_3,n}\bar{h}_{I_3D_1,n}\eta_{3n}$, $\eta_{3n} = \eta, \forall_{3n}$.

In (14), to obtain the maximum value of $\gamma_{D_1}^{(2)}$, we need to eliminate the channel phases. Similar to the method in [35], by setting the phases $\theta_{3n}^{(D_1)} = \phi_{3n} + \delta_{3n}^{(D_1)}$ for $3n = 1, \ldots, N_3$, the maximal $\gamma_{D_1}^{(2)}$ can be written as

$$\gamma_{D_1}^{(2)} = \frac{\gamma\chi_1\left|A_1^{(1)} + A_2^{(1)} + A_3^{(1)}\right|^2}{\gamma\chi_2\left|A_1^{(1)} + A_2^{(1)} + A_3^{(1)}\right|^2 + \left(Y_S^2 + Y_{D_1}^2\right)\gamma\left|A_1^{(1)} + A_2^{(1)} + A_3^{(1)}\right|^2 + 1}. \tag{15}$$

The overall received signal at the user $D_2$ is given as [33,36]

$$y_{D_2}^{(2)} = \left(\sum_{n=1}^{N_1} h_{SI_1,n} h_{I_1D_2,n} \eta_{1n} e^{j\theta_{1n}^{(D_2)}} + \sum_{n=1}^{N_2} h_{SI_2,n} h_{I_2D_2,n} \eta_{2n} e^{j\theta_{2n}^{(D_2)}} + \sum_{n=1}^{N_3} h_{SI_3,n} h_{I_3D_2,n} \eta_{3n} e^{j\theta_{3n}^{(D_2)}}\right)\left(\sqrt{P\chi_1}x_1 + \sqrt{P\chi_2}x_2 + \tau_S\right) \tag{16}$$
$$+ \tau_{D_{32}} + n_{D_2},$$

where $\tau_{D_{32}} \sim CN\left(0, Y_{D_2}^2 P\left|\sum_{n=1}^{N_1} h_{SI_1,n} h_{I_1D_2,n} \eta_{1n} e^{j\theta_{1n}^{(D_2)}} + \sum_{n=1}^{N_2} h_{SI_2,n} h_{I_2D_2,n} \eta_{2n} e^{j\theta_{2n}^{(D_2)}}\right.\right.$

$\left.\left.+ \sum_{n=1}^{N_3} h_{SI_3,n} h_{I_3D_2,n} \eta_{3n} e^{j\theta_{3n}^{(D_2)}}\right|^2\right)$. $\theta_{3n}^{(D_2)}$ is the adjustable phase applied by the $n$th reflecting element of $S - I_3 - D_2$. In addition, $h_{I_3D_2,n}$ is the channel gains with $h_{I_3D_2,n} = d_{I_3D_2}^{-\alpha/2}\bar{h}_{I_3D_2,n}e^{-j\delta_{3n}^{(D_2)}}$ [35], where $d_{I_3D_2}$ is the distance for the $I_3 - D_2$ link, and

$\delta_{3n}^{(D_2)}$ is the phase of the channel gains. $\bar{h}_{I_3 D_2,n}$ following a Rayleigh distribution is the amplitude of the channel gains. Moreover, we assume that the IRS has full knowledge of the channel phases of $h_{I_3 D_2,n}$.

Similar to (9), the resulting SNDR at the legitimate user $D_2$ to decode $x_1$ can be formulated as

$$\gamma_{x_1,D_2}^{(2)} = \frac{\gamma \chi_1 \left| B_1^{(1)} + B_2^{(1)} + B_3^{(1)} \right|^2}{\gamma \chi_2 \left| B_1^{(1)} + B_2^{(1)} + B_3^{(1)} \right|^2 + \left( Y_S^2 + Y_{D_2}^2 \right) \gamma \left| B_1^{(1)} + B_2^{(1)} + B_3^{(1)} \right|^2 + 1}, \tag{17}$$

where $B_3^{(1)} = d_{SI_3}^{-\alpha/2} d_{I_3 D_2}^{-\alpha/2} \eta \sum_{n=1}^{N_3} \bar{h}_{SI_3,n} \bar{h}_{I_3 D_2,n}$.

To obtain the maximum value of $\gamma_{x_1,D_2}^{(2)}$, we need to eliminate the channel phases. Similar to the method in [35], by setting the phases $\theta_{3n}^{(D_2)} = \phi_{3n} + \delta_{3n}^{(D_2)}$ for $3n = 1, \dots, N_3$.

After SIC, the resulting SNDR at the user $D_2$ to decode $x_2$ can be formulated as

$$\gamma_{x_2,D_2}^{(2)} = \frac{\gamma \chi_2 \left| B_1^{(1)} + B_2^{(1)} + B_3^{(1)} \right|^2}{\left( Y_S^2 + Y_{D_2}^2 \right) \gamma \left| B_1^{(1)} + B_2^{(1)} + B_3^{(1)} \right|^2 + 1}. \tag{18}$$

## 3. Ergodic Performance Analysis of the Proposed Scheme Using Rayleigh and Rician Fading Channels

In this section, two types of fading channels are considered for $S \rightarrow I_1$: $I_1 \rightarrow D_i$, $S \rightarrow I_2$, $I_2 \rightarrow D_i$, $S \rightarrow I_3$, and $I_3 \rightarrow D_i$, i.e., Rayleigh and Rician fading channels (similar to [31], these channel distributions are preferred to examine ergodic performance, while the Nakagami-$m$ demonstrated similar performance, and hence we do not want to evaluate the case of Nakagami-$m$ in the scope of this study). In the following sections, upper bounds using Rayleigh and Rician fading channels are derived for our proposed scheme.

### 3.1. The First Scenario

3.1.1. Upper Bound for the Achievable Rate Using Rayleigh Fading Channels for $D_1$

In the case of Rayleigh fading channels, the existing links $S - I_1 - D_i$, $S - I_2 - D_i$ and $S - I_3 - D_i$ are assumed to have only non-line of sight (NLOS) components and hence can be modeled as Rayleigh fading channels, i.e., $h_l = \bar{h}_l d_l^{-\alpha/2}$, $l \in \{SI_1, I_1 D_i, SI_2, I_2 D_i, SI_3, I_3 D_i\}$, where $\bar{h}_l$ is the complex-Gaussian small-scale fading channels with zero mean and unit variance.

**Proposition 1.** *The upper bound for $D_1$ using the Rayleigh fading channels is given as*

$$\begin{aligned} R_{\text{upper},D_1}^{(1)} &= \log_2 \left( 1 + E \left[ \widetilde{\gamma_{D_1}^{(1)}} \right] \right) \\ &= \log_2 \left( 1 + \frac{\gamma \chi_1 \left( \frac{1}{16}\beta_1 + \frac{1}{8}\beta_2 + \frac{1}{16}\beta_3 \right)}{\gamma \chi_2 \left( \frac{1}{16}\beta_1 + \frac{1}{8}\beta_2 + \frac{1}{16}\beta_3 \right) + \left( Y_S^2 + Y_{D_1}^2 \right) \gamma \left( \frac{1}{16}\beta_1 + \frac{1}{8}\beta_2 + \frac{1}{16}\beta_3 \right) + 1} \right), \end{aligned} \tag{19}$$

*where $\beta_1 = d_{SI_1}^{-\alpha} d_{I_1 D_1}^{-\alpha} \eta^2 N_1 \left( 16 + (N_1 - 1)\pi^2 \right)$, $\beta_2 = d_{SI_1}^{-\alpha/2} d_{I_1 D_1}^{-\alpha/2} d_{SI_2}^{-\alpha/2} d_{I_2 D_1}^{-\alpha/2} \eta \eta$ $\sqrt{N_1 N_2 (16 + (N_1 - 1)\pi^2)(16 + (N_2 - 1)\pi^2)}$, $\beta_3 = d_{SI_2}^{-\alpha} d_{I_2 D_1}^{-\alpha} \eta^2 N_2 \left( 16 + (N_2 - 1)\pi^2 \right)$.*

**Proof.** See Appendix A. □

3.1.2. Upper Bound for the Achievable Rate Using Rayleigh Fading Channels for $D_2$

**Proposition 2.** *The upper bound for the $D_2$ using the Rayleigh fading channels is given as*

$$R^{(1)}_{\text{upper},D_2} = \log_2\left(1 + \min\left\{E\left[\widetilde{\gamma^{(1)}_{x_1,D_2}}\right], E\left[\widetilde{\gamma^{(1)}_{x_2,D_2}}\right]\right\}\right)$$

$$= \log_2\left(1 + \min\left\{\frac{\gamma\chi_1\left(\frac{1}{16}\psi_1 + \frac{1}{8}\psi_2 + \frac{1}{16}\psi_3\right)}{\gamma\chi_2\left(\frac{1}{16}\psi_1 + \frac{1}{8}\psi_2 + \frac{1}{16}\psi_3\right) + \left(Y_S^2 + Y_{D_2}^2\right)\gamma\left(\frac{1}{16}\psi_1 + \frac{1}{8}\psi_2 + \frac{1}{16}\psi_3\right) + 1}, \frac{\gamma\chi_2\left(\frac{1}{16}\psi_1 + \frac{1}{8}\psi_2 + \frac{1}{16}\psi_3\right)}{\left(Y_S^2 + Y_{D_2}^2\right)\gamma\left(\frac{1}{16}\psi_1 + \frac{1}{8}\psi_2 + \frac{1}{16}\psi_3\right) + 1}\right\}\right),$$

(20)

*where* $\psi_1 = d_{SI_1}^{-\alpha} d_{I_1 D_2}^{-\alpha} \eta^2 N_1 \left(16 + (N_1 - 1)\pi^2\right)$, $\psi_2 = d_{SI_1}^{-\alpha/2} d_{I_1 D_2}^{-\alpha/2} d_{SI_2}^{-\alpha/2} d_{I_2 D_2}^{-\alpha/2} \eta\eta$
$\sqrt{N_1 N_2 \left(16 + (N_1 - 1)\pi^2\right)\left(16 + (N_2 - 1)\pi^2\right)}$, $\psi_3 = d_{SI_2}^{-\alpha} d_{I_2 D_2}^{-\alpha} \eta^2 N_2 \left(16 + (N_2 - 1)\pi^2\right)$.

**Proof.** See Appendix B. □

3.1.3. Upper Bound for the Achievable Rate Using Rician Fading Channels for $D_1$

In this case, it is assumed that line-of-sight (LOS) paths are presented between the links in $SI_1 D_i$, $SI_2 D_i$ and $SI_3 D_i$ and are modeled as Rician fading channels, $h_p = \sqrt{\frac{K_p}{K_p + 1}} \hat{h}_p d_p^{-\alpha/2}, p \in \{SI_1, I_1 D_i, SI_2, I_2 D_i, SI_3, I_3 D_i\}$, where $\hat{h}_p$ is a fixed-component vector with elements of unit power, and $K_p$ is the Rician K-factor.

**Proposition 3.** *The upper bound for the $D_1$ using the Rician fading channels is given as*

$$\hat{R}^{(1)}_{\text{upper},D_1} = \log_2\left(1 + E\left[\widetilde{\gamma^{(1)}_{D_1}}\right]\right)$$

$$= \log_2\left(1 + \frac{\gamma\chi_1(\partial_1 + 2\partial_2 + \partial_3)}{\gamma\chi_2(\partial_1 + 2\partial_2 + \partial_3) + \left(Y_S^2 + Y_{D_1}^2\right)\gamma(\partial_1 + 2\partial_2 + \partial_3) + 1}\right),$$

(21)

*where* $\partial_1 = \frac{\eta^2 N_1^2 \rho_{SI_1 D_1}^2}{d_{SI_1}^\alpha d_{I_1 D_1}^\alpha}$, $\partial_2 = \frac{\eta\eta N_1 N_2 \rho_{SI_1 D_1} \rho_{SI_2 D_1}}{d_{SI_1}^{\alpha/2} d_{I_1 D_1}^{\alpha/2} d_{SI_2}^{\alpha/2} d_{I_2 D_1}^{\alpha/2}}$, $\partial_3 = \frac{\eta^2 N_2^2 \rho_{SI_2 D_1}^2}{d_{SI_2}^\alpha d_{I_2 D_1}^\alpha}$.

**Proof.** See Appendix C. □

3.1.4. Upper Bound for the Achievable Rate Using Rician Fading Channels for $D_2$

**Proposition 4.** *The upper bound for the $D_2$ using the Rician fading channels is given as*

$$\hat{R}^{(1)}_{\text{upper},D_2} = \log_2\left(1 + \min\left\{E\left[\widetilde{\gamma^{(1)}_{x_1,D_2}}\right], E\left[\widetilde{\gamma^{(1)}_{x_2,D_2}}\right]\right\}\right)$$

$$= \log_2\left(1 + \min\left\{\frac{\gamma\chi_1(\omega_1 + 2\omega_2 + \omega_3)}{\gamma\chi_2(\omega_1 + 2\omega_2 + \omega_3) + \left(Y_S^2 + Y_{D_2}^2\right)\gamma(\omega_1 + 2\omega_2 + \omega_3) + 1}, \frac{\gamma\chi_2(\omega_1 + 2\omega_2 + \omega_3)}{\left(Y_S^2 + Y_{D_2}^2\right)\gamma(\omega_1 + 2\omega_2 + \omega_3) + 1}\right\}\right),$$

(22)

*where* $\omega_1 = \frac{\eta^2 N_1^2 \rho_{SI_1 D_2}^2}{d_{SI_1}^\alpha d_{I_1 D_2}^\alpha}$, $\omega_2 = \frac{\eta\eta N_1 N_2 \rho_{SI_1 D_2} \rho_{SI_2 D_2}}{d_{SI_1}^{\alpha/2} d_{I_1 D_2}^{\alpha/2} d_{SI_2}^{\alpha/2} d_{I_2 D_2}^{\alpha/2}}$, $\omega_3 = \frac{\eta^2 N_2^2 \rho_{SI_2 D_2}^2}{d_{SI_2}^\alpha d_{I_2 D_2}^\alpha}$.

**Proof.** See Appendix D. □

*3.2. The Second Scenario (the Benchmark)*

3.2.1. Upper Bound for the Achievable Rate Using Rayleigh Fading Channels for $D_1$

Let us denote $\sqrt{\widetilde{A_3^{(1)}}} = \frac{\eta}{N_3} d_{SI_3}^{-\alpha/2} d_{I_3 D_1}^{-\alpha/2} \sum_{n=1}^{N_3} \bar{h}_{SI_3,n} \bar{h}_{I_3 D_1,n}$. It is noted that $\widetilde{A_3^{(1)}}$ follows a non-central chi-square distribution with mean values given as [36]

$$E\left[\widetilde{A_3^{(1)}}\right] = \frac{\eta^2\left[\pi^2 + (1/N_3)(16 - \pi^2)\right]}{16 d_{SI_3}^\alpha d_{I_3 D_1}^\alpha}.$$

(23)

Therefore, the expected value of $\gamma_{D_1}^{(2)}$ can be derived as

$$\mathrm{E}\left[\widetilde{\gamma_{D_1}^{(2)}}\right] = \mathrm{E}\left[\frac{\gamma\chi_1\left(N_1\sqrt{\widetilde{A_1^{(1)}}}+N_2\sqrt{\widetilde{A_2^{(1)}}}+N_3\sqrt{\widetilde{A_3^{(1)}}}\right)^2}{\gamma\chi_2\left(N_1\sqrt{\widetilde{A_1^{(1)}}}+N_2\sqrt{\widetilde{A_2^{(1)}}}+N_3\sqrt{\widetilde{A_3^{(1)}}}\right)^2+\left(Y_5^2+Y_{D_1}^2\right)\gamma\left(N_1\sqrt{\widetilde{A_1^{(1)}}}+N_2\sqrt{\widetilde{A_2^{(1)}}}+N_3\sqrt{\widetilde{A_3^{(1)}}}\right)^2+1}\right]$$

$$= \frac{\gamma\chi_1\Gamma_1}{\gamma\chi_2\Gamma_1+\left(Y_5^2+Y_{D_1}^2\right)\gamma\Gamma_1+1}$$

$$\leq \frac{\gamma\chi_1\left(\frac{1}{16}\beta_1+\frac{1}{16}\beta_3+\frac{1}{16}\beta_4+\frac{1}{8}\beta_2+\frac{1}{8}\beta_5+\frac{1}{8}\beta_6\right)}{\gamma\chi_2\left(\frac{1}{16}\beta_1+\frac{1}{16}\beta_3+\frac{1}{16}\beta_4+\frac{1}{8}\beta_2+\frac{1}{8}\beta_5+\frac{1}{8}\beta_6\right)+\left(Y_5^2+Y_{D_1}^2\right)\gamma\left(\frac{1}{16}\beta_1+\frac{1}{16}\beta_3+\frac{1}{16}\beta_4+\frac{1}{8}\beta_2+\frac{1}{8}\beta_5+\frac{1}{8}\beta_6\right)+1},$$

(24)

in which $\Gamma_1 = N_1^2\mathrm{E}\left[\widetilde{A_1^{(1)}}\right] + 2N_1N_2\mathrm{E}\left[\sqrt{\widetilde{A_1^{(1)}}}\sqrt{\widetilde{A_2^{(1)}}}\right] + N_2^2\mathrm{E}\left[\widetilde{A_2^{(1)}}\right] + N_3^2\mathrm{E}\left[\widetilde{A_3^{(1)}}\right]$

$$+ \qquad 2N_1N_3\mathrm{E}\left[\sqrt{\widetilde{A_1^{(1)}}}\sqrt{\widetilde{A_3^{(1)}}}\right] \qquad + \qquad 2N_2N_3\mathrm{E}\left[\sqrt{\widetilde{A_2^{(1)}}}\sqrt{\widetilde{A_3^{(1)}}}\right],$$

$$\beta_4 = d_{SI_3}^{-\alpha}d_{I_3D_1}^{-\alpha}\eta^2N_3\left(16+(N_3-1)\pi^2\right),$$

$$\beta_5 = d_{SI_1}^{-\alpha/2}d_{I_1D_1}^{-\alpha/2}d_{SI_3}^{-\alpha/2}d_{I_3D_1}^{-\alpha/2}\eta\eta\sqrt{N_1N_3[16+(N_1-1)\pi^2][16+(N_3-1)\pi^2]},$$

$$\beta_6 = d_{SI_2}^{-\alpha/2}d_{I_2D_1}^{-\alpha/2}d_{SI_3}^{-\alpha/2}d_{I_3D_1}^{-\alpha/2}\eta\eta\sqrt{N_2N_3[16+(N_2-1)\pi^2][16+(N_3-1)\pi^2]}.$$

The upper bound for the $D_1$ using the Rayleigh fading channels is given as

$$R_{\mathrm{upper},D_1}^{(2)} = \log_2\left(1+\mathrm{E}\left[\widetilde{\gamma_{D_1}^{(2)}}\right]\right).$$

(25)

### 3.2.2. Upper Bound for the Achievable Rate Using Rayleigh Fading Channels for $D_2$

We denote $\sqrt{\widetilde{B_3^{(1)}}} = \frac{\eta}{N_3}d_{SI_3}^{-\alpha/2}d_{I_3D_2}^{-\alpha/2}\sum_{n=1}^{N_3}\bar{h}_{SI_3,n}\bar{h}_{I_3D_2,n}$ [36], while $\widetilde{B_3^{(1)}}$ follows a non-central chi-square distribution with mean values given as [36]

$$\mathrm{E}\left[\widetilde{B_3^{(1)}}\right] = \frac{\eta^2\left[\pi^2+(1/N_3)(16-\pi^2)\right]}{16d_{SI_3}^{\alpha}d_{I_3D_2}^{\alpha}}.$$

(26)

The expected value of $\gamma_{x_1,D_2}^{(2)}$ can be derived as

$$\mathrm{E}\left[\widetilde{\gamma_{x_1,D_2}^{(2)}}\right] = \mathrm{E}\left[\frac{\gamma\chi_1\left(N_1\sqrt{\widetilde{B_1^{(1)}}}+N_2\sqrt{\widetilde{B_2^{(1)}}}+N_3\sqrt{\widetilde{B_3^{(1)}}}\right)^2}{\gamma\chi_2\left(N_1\sqrt{\widetilde{B_1^{(1)}}}+N_2\sqrt{\widetilde{B_2^{(1)}}}+N_3\sqrt{\widetilde{B_3^{(1)}}}\right)^2+\left(Y_5^2+Y_{D_2}^2\right)\gamma\left(N_1\sqrt{\widetilde{B_1^{(1)}}}+N_2\sqrt{\widetilde{B_2^{(1)}}}+N_3\sqrt{\widetilde{B_3^{(1)}}}\right)^2+1}\right]$$

$$= \frac{\gamma\chi_1\Gamma_2}{\gamma\chi_2\Gamma_2+\left(Y_5^2+Y_{D_2}^2\right)\gamma\Gamma_2+1}$$

$$\leq \frac{\gamma\chi_1\left(\frac{1}{16}\psi_1+\frac{1}{16}\psi_3+\frac{1}{16}\psi_4+\frac{1}{8}\psi_2+\frac{1}{8}\psi_5+\frac{1}{8}\psi_6\right)}{\gamma\chi_2\left(\frac{1}{16}\psi_1+\frac{1}{16}\psi_3+\frac{1}{16}\psi_4+\frac{1}{8}\psi_2+\frac{1}{8}\psi_5+\frac{1}{8}\psi_6\right)+\left(Y_5^2+Y_{D_2}^2\right)\gamma\left(\frac{1}{16}\psi_1+\frac{1}{16}\psi_3+\frac{1}{16}\psi_4+\frac{1}{8}\psi_2+\frac{1}{8}\psi_5+\frac{1}{8}\psi_6\right)+1},$$

(27)

where $\Gamma_2 = N_1^2\mathrm{E}\left[\widetilde{B_1^{(1)}}\right] + 2N_1N_2\mathrm{E}\left[\sqrt{\widetilde{B_1^{(1)}}}\sqrt{\widetilde{B_2^{(1)}}}\right] + N_2^2\mathrm{E}\left[\widetilde{B_2^{(1)}}\right] + N_3^2\mathrm{E}\left[\widetilde{B_3^{(1)}}\right] + 2N_1N_3$

$$\mathrm{E}\left[\sqrt{\widetilde{B_1^{(1)}}}\sqrt{\widetilde{B_3^{(1)}}}\right] + 2N_2N_3\mathrm{E}\left[\sqrt{\widetilde{B_2^{(1)}}}\sqrt{\widetilde{B_3^{(1)}}}\right], \quad \psi_4 = d_{SI_3}^{-\alpha}d_{I_3D_2}^{-\alpha}\eta^2N_3\left[16+(N_3-1)\pi^2\right],$$

$$\psi_5 = d_{SI_1}^{-\alpha/2}d_{I_1D_2}^{\alpha/2}d_{SI_3}^{-\alpha/2}d_{I_3D_2}^{-\alpha/2}\eta\eta\sqrt{N_1N_3[16+(N_1-1)\pi^2][16+(N_3-1)\pi^2]},$$

$$\psi_6 = d_{SI_2}^{-\alpha/2}d_{I_2D_2}^{-\alpha/2}d_{SI_3}^{-\alpha/2}d_{I_3D_2}^{-\alpha/2}\eta\eta\sqrt{N_2N_3[16+(N_2-1)\pi^2][16+(N_3-1)\pi^2]}.$$

The expected value of $\gamma_{x_2,D_2}^{(2)}$ can be derived as

$$
\begin{aligned}
\mathrm{E}\left[\widetilde{\gamma_{x_2,D_2}^{(2)}}\right] &= \mathrm{E}\left[\frac{\gamma\chi_2\left(N_1\sqrt{\widetilde{B_1^{(1)}}}+N_2\sqrt{\widetilde{B_2^{(1)}}}+N_3\sqrt{\widetilde{B_3^{(1)}}}\right)^2}{\left(\mathrm{Y}_S^2+\mathrm{Y}_{D_2}^2\right)\gamma\left(N_1\sqrt{\widetilde{B_1^{(1)}}}+N_2\sqrt{\widetilde{B_2^{(1)}}}+N_3\sqrt{\widetilde{B_3^{(1)}}}\right)^2+1}\right] \\
&= \frac{\gamma\chi_2\Gamma_2}{\left(\mathrm{Y}_S^2+\mathrm{Y}_{D_2}^2\right)\gamma\Gamma_2+1} \\
&\leq \frac{\gamma\chi_2\left(\frac{1}{16}\psi_1+\frac{1}{16}\psi_3+\frac{1}{16}\psi_4+\frac{1}{8}\psi_2+\frac{1}{8}\psi_5+\frac{1}{8}\psi_6\right)}{\left(\mathrm{Y}_S^2+\mathrm{Y}_{D_2}^2\right)\gamma\left(\frac{1}{16}\psi_1+\frac{1}{16}\psi_3+\frac{1}{16}\psi_4+\frac{1}{8}\psi_2+\frac{1}{8}\psi_5+\frac{1}{8}\psi_6\right)+1}.
\end{aligned}
\tag{28}
$$

The upper bound for the $D_2$ using the Rayleigh fading channels is given as

$$
R_{\mathrm{upper},D_2}^{(2)} = \log_2\left(1+\min\left\{\mathrm{E}\left[\widetilde{\gamma_{x_1,D_2}^{(2)}}\right],\mathrm{E}\left[\widetilde{\gamma_{x_2,D_2}^{(2)}}\right]\right\}\right).
\tag{29}
$$

**Remark 1.** *These results of ergodic performance analysis are necessary to determine how many IRSs are sufficient to maintain the performance of IoT devices. For example,* (28) *gives us the hardware impairment levels and settings of IRSs playing the main role in the performance variations. We expect to evaluate more parameters in the numerical simulation section.*

### 3.2.3. Upper Bound for the Achievable Rate Using Rician Fading Channels for $D_1$

In this case, we denote $\sqrt{\widetilde{A_3^{(1)}}} = \frac{\eta}{N_3}\rho_{SI_3D_1}d_{SI_3}^{-\alpha/2}d_{I_3D_1}^{-\alpha/2}\sum\limits_{n=1}^{N_3}\hat{h}_{SI_3,n}\hat{h}_{I_3D_1,n}$ where $\rho_{SI_3D_1} = \sqrt{\frac{K_{SI_3}K_{I_3D_1}}{(K_{SI_3}+1)(K_{I_3D_1}+1)}}$. Moreover, $\widetilde{A_3^{(1)}}$ has constant mean values of $\mathrm{E}\left[\widetilde{A_3^{(1)}}\right] = \frac{\eta^2\rho_{SI_3D_1}^2}{d_{SI_3}^\alpha d_{I_3D_1}^\alpha}$ [36].

Therefore, the expected value of $\gamma_{D_1}^{(2)}$ can be derived as

$$
\begin{aligned}
\mathrm{E}\left[\widetilde{\gamma_{D_1}^{(2)}}\right] &= \mathrm{E}\left[\frac{\gamma\chi_1\left(N_1\sqrt{\widetilde{A_1^{(1)}}}+N_2\sqrt{\widetilde{A_2^{(1)}}}+N_3\sqrt{\widetilde{A_3^{(1)}}}\right)^2}{\gamma\chi_2\left(N_1\sqrt{\widetilde{A_1^{(1)}}}+N_2\sqrt{\widetilde{A_2^{(1)}}}+N_3\sqrt{\widetilde{A_3^{(1)}}}\right)^2+\left(\mathrm{Y}_S^2+\mathrm{Y}_{D_1}^2\right)\gamma\left(N_1\sqrt{\widetilde{A_1^{(1)}}}+N_2\sqrt{\widetilde{A_2^{(1)}}}+N_3\sqrt{\widetilde{A_3^{(1)}}}\right)^2+1}\right] \\
&= \frac{\gamma\chi_1\Gamma_3}{\gamma\chi_2\Gamma_3+\left(\mathrm{Y}_S^2+\mathrm{Y}_{D_1}^2\right)\gamma\Gamma_3+1} \\
&\leq \frac{\gamma\chi_1(\partial_1+\partial_3+\partial_4+2\partial_2+2\partial_5+2\partial_6)}{\gamma\chi_2(\partial_1+\partial_3+\partial_4+2\partial_2+2\partial_5+2\partial_6)+\left(\mathrm{Y}_S^2+\mathrm{Y}_{D_1}^2\right)\gamma(\partial_1+\partial_3+\partial_4+2\partial_2+2\partial_5+2\partial_6)+1},
\end{aligned}
\tag{30}
$$

in which $\Gamma_3 = N_1^2\mathrm{E}\left[\widetilde{A_1^{(1)}}\right]+2N_1N_2\mathrm{E}\left[\sqrt{\widetilde{A_1^{(1)}}}\sqrt{\widetilde{A_2^{(1)}}}\right]+N_2^2\mathrm{E}\left[\widetilde{A_2^{(1)}}\right]+N_3^2\mathrm{E}\left[\widetilde{A_3^{(1)}}\right]+2N_1N_3$

$\mathrm{E}\left[\sqrt{\widetilde{A_1^{(1)}}}\sqrt{\widetilde{A_3^{(1)}}}\right]+2N_2N_3\mathrm{E}\left[\sqrt{\widetilde{A_2^{(1)}}}\sqrt{\widetilde{A_3^{(1)}}}\right]$, $\partial_4 = \frac{\eta^2N_3^2\rho_{SI_3D_1}^2}{d_{SI_3}^\alpha d_{I_3D_1}^\alpha}$, $\partial_5 = \frac{\eta\eta N_1N_3\rho_{SI_1D_1}\rho_{SI_3D_1}}{d_{SI_1}^{\alpha/2}d_{I_1D_1}^{\alpha/2}d_{SI_3}^{\alpha/2}d_{I_3D_1}^{\alpha/2}}$,

$\partial_6 = \frac{\eta\eta N_2N_3\rho_{SI_2D_1}\rho_{SI_3D_1}}{d_{SI_2}^{\alpha/2}d_{I_2D_1}^{\alpha/2}d_{SI_3}^{\alpha/2}d_{I_3D_1}^{\alpha/2}}$.

The upper bound for the $D_1$ using the Rician fading channels is given as

$$
\widehat{R}_{\mathrm{upper},D_1}^{(2)} = \log_2\left(1+\mathrm{E}\left[\widetilde{\gamma_{D_1}^{(2)}}\right]\right).
\tag{31}
$$

### 3.2.4. Upper Bound for the Achievable Rate Using Rician Fading Channels for $D_2$

Now, defining $\sqrt{\widehat{B_3^{(1)}}} = \frac{\eta}{N_3} \rho_{SI_3D_2} d_{SI_3}^{-\alpha/2} d_{I_3D_2}^{-\alpha/2} \sum\limits_{n=1}^{N_3} \hat{h}_{SI_3,n} \hat{h}_{I_3D_2,n}$, where $\rho_{SI_3D_2} =$
$\sqrt{\frac{K_{SI_3} K_{I_3D_2}}{(K_{SI_3}+1)(K_{I_3D_2}+1)}}$. Moreover, $\widehat{B_3^{(1)}}$ has constant mean values of $E\left[\widehat{B_3^{(1)}}\right] = \frac{\eta^2 \rho_{SI_3D_2}^2}{d_{SI_3}^\alpha d_{I_3D_2}^\alpha}$ [36].

The expected value of $\gamma_{x_1,D_2}^{(2)}$ can be derived as

$$
\begin{aligned}
E\left[\widehat{\gamma_{x_1,D_2}^{(2)}}\right] &= E\left[\frac{\gamma\chi_1\left(N_1\sqrt{\widehat{B_1^{(1)}}}+N_2\sqrt{\widehat{B_2^{(1)}}}+N_3\sqrt{\widehat{B_3^{(1)}}}\right)^2}{\gamma\chi_2\left(N_1\sqrt{\widehat{B_1^{(1)}}}+N_2\sqrt{\widehat{B_2^{(1)}}}+N_3\sqrt{\widehat{B_3^{(1)}}}\right)^2+\left(Y_S^2+Y_{D_2}^2\right)\gamma\left(N_1\sqrt{\widehat{B_1^{(1)}}}+N_2\sqrt{\widehat{B_2^{(1)}}}+N_3\sqrt{\widehat{B_3^{(1)}}}\right)^2+1}\right] \\
&= \frac{\gamma\chi_1\Gamma_4}{\gamma\chi_2\Gamma_4+\left(Y_S^2+Y_{D_2}^2\right)\gamma\Gamma_4+1} \\
&\leq \frac{\gamma\chi_1(\omega_1+\omega_3+\omega_4+2\omega_2+2\omega_5+2\omega_6)}{\gamma\chi_2(\omega_1+\omega_3+\omega_4+2\omega_2+2\omega_5+2\omega_6)+\left(Y_S^2+Y_{D_2}^2\right)\gamma(\omega_1+\omega_3+\omega_4+2\omega_2+2\omega_5+2\omega_6)+1},
\end{aligned}
\tag{32}
$$

where $\Gamma_4 = N_1^2 E\left[\widehat{B_1^{(1)}}\right] + 2N_1N_2 E\left[\sqrt{\widehat{B_1^{(1)}}}\sqrt{\widehat{B_2^{(1)}}}\right] + N_2^2 E\left[\widehat{B_2^{(1)}}\right] + N_3^2 E\left[\widehat{B_3^{(1)}}\right] + 2N_1N_3$

$E\left[\sqrt{\widehat{B_1^{(1)}}}\sqrt{\widehat{B_3^{(1)}}}\right] + 2N_2N_3 E\left[\sqrt{\widehat{B_2^{(1)}}}\sqrt{\widehat{B_3^{(1)}}}\right]$, $\omega_4 = \frac{\eta^2 N_3^2 \rho_{SI_3D_2}^2}{d_{SI_3}^\alpha d_{I_3D_2}^\alpha}$, $\omega_5 = \frac{\eta\eta N_1N_3 \rho_{SI_1D_2}\rho_{SI_3D_2}}{d_{SI_1}^{\alpha/2} d_{I_1D_2}^{\alpha/2} d_{SI_3}^{\alpha/2} d_{I_3D_2}^{\alpha/2}}$,

$\omega_6 = \frac{\eta\eta N_2N_3 \rho_{SI_2D_2}\rho_{SI_3D_2}}{d_{SI_2}^{\alpha/2} d_{I_2D_2}^{\alpha/2} d_{SI_3}^{\alpha/2} d_{I_3D_2}^{\alpha/2}}$.

The expected value of $\gamma_{x_2,D_2}^{(2)}$ can be derived as

$$
\begin{aligned}
E\left[\widehat{\gamma_{x_2,D_2}^{(2)}}\right] &= E\left[\frac{\gamma\chi_2\left(N_1\sqrt{\widehat{B_1^{(1)}}}+N_2\sqrt{\widehat{B_2^{(1)}}}+N_3\sqrt{\widehat{B_3^{(1)}}}\right)^2}{\left(Y_S^2+Y_{D_2}^2\right)\gamma\left(N_1\sqrt{\widehat{B_1^{(1)}}}+N_2\sqrt{\widehat{B_2^{(1)}}}+N_3\sqrt{\widehat{B_3^{(1)}}}\right)^2+1}\right] \\
&= \frac{\gamma\chi_2\Gamma_4}{\left(Y_S^2+Y_{D_2}^2\right)\gamma\Gamma_4+1} \\
&\leq \frac{\gamma\chi_2(\omega_1+\omega_3+\omega_4+2\omega_2+2\omega_5+2\omega_6)}{\left(Y_S^2+Y_{D_2}^2\right)\gamma(\omega_1+\omega_3+\omega_4+2\omega_2+2\omega_5+2\omega_6)+1}.
\end{aligned}
\tag{33}
$$

The upper bound for the $D_2$ using the Rician fading channels is given as

$$
\widehat{R}_{\text{upper},D_2}^{(2)} = \log_2\left(1+\min\left\{E\left[\widehat{\gamma_{x_1,D_2}^{(2)}}\right], E\left[\widehat{\gamma_{x_2,D_2}^{(2)}}\right]\right\}\right).
\tag{34}
$$

**Remark 2.** *It is difficult to determine how channel models (Rayleigh and Rician) affect the performance of IoT devices. For example, (34) gives us K Rician factors, and the setting IRSs still plays a main role in the performance variations. We determine more parameters in the numerical simulation section.*

## 4. Simulation Results

In this section, simulation results are provided to evaluate and assess the capacity performance of the proposed scheme of using multiple IRSs for the IRS–NOMA–UAV system. The single antenna source is placed at the origin $(x_S, y_S) = (0,0)$, the destinations at $(x_{D_1}, y_{D_1}) = (100,0)$, $(x_{D_2}, y_{D_2}) = (90,0)$, and the three IRSs at $(x_{I_1}, y_{I_1}) = (40,10)$, $(x_{I_2}, y_{I_2}) = (50,10)$, $(x_{I_3}, y_{I_3}) = (60,10)$ [36]. The power allocation factor $\chi_1 = 0.6$ [34], the path-loss $\alpha = 2$ [34], SNR $\gamma = 30$ (dB) and reflection coefficients $\eta = 0.7$, $N_1 = N_2 = N_3 = 200$ [30,33], while the Rician-$K$ factor for all links from and to the IRS was 10 (dB) [36], $Y_S = Y_{D_1} = Y_{D_2} = 0.05$ [32]. We adopt the principle of Monte Carlo simulations as shown in Figure 2.

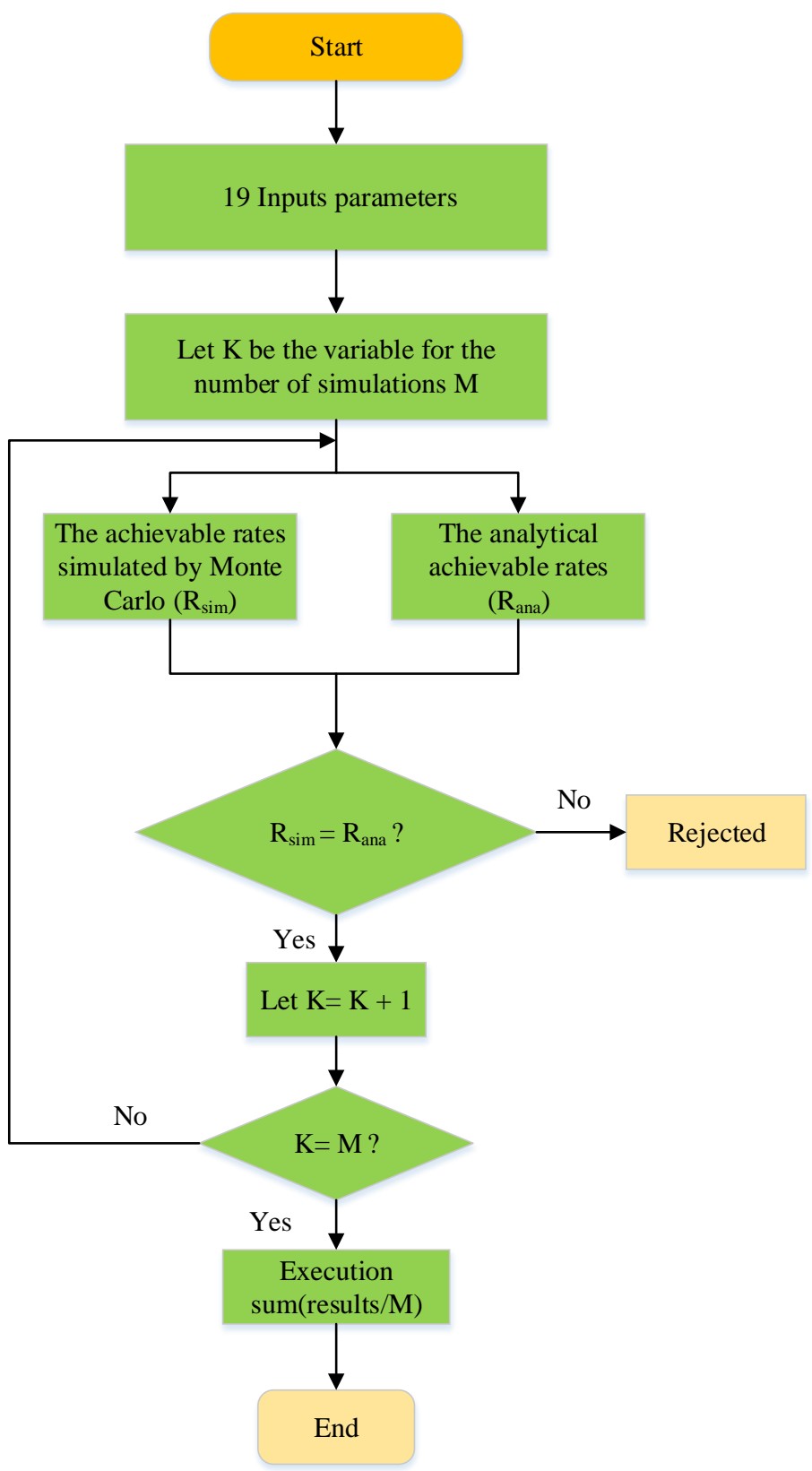

**Figure 2.** The flowchart of Monte Carlo simulations.

Figure 3 demonstrates the ergodic capacity performance as a function of $N$ and $\gamma$ for two scenarios. When the IRS elements give additional constructive paths, an enhancement in the SNDR could be affected by the increasing $N$. It is intuitively seen that a higher average SNR at the source leads to a remarkable gap between the ergodic capacity of the two scenarios.

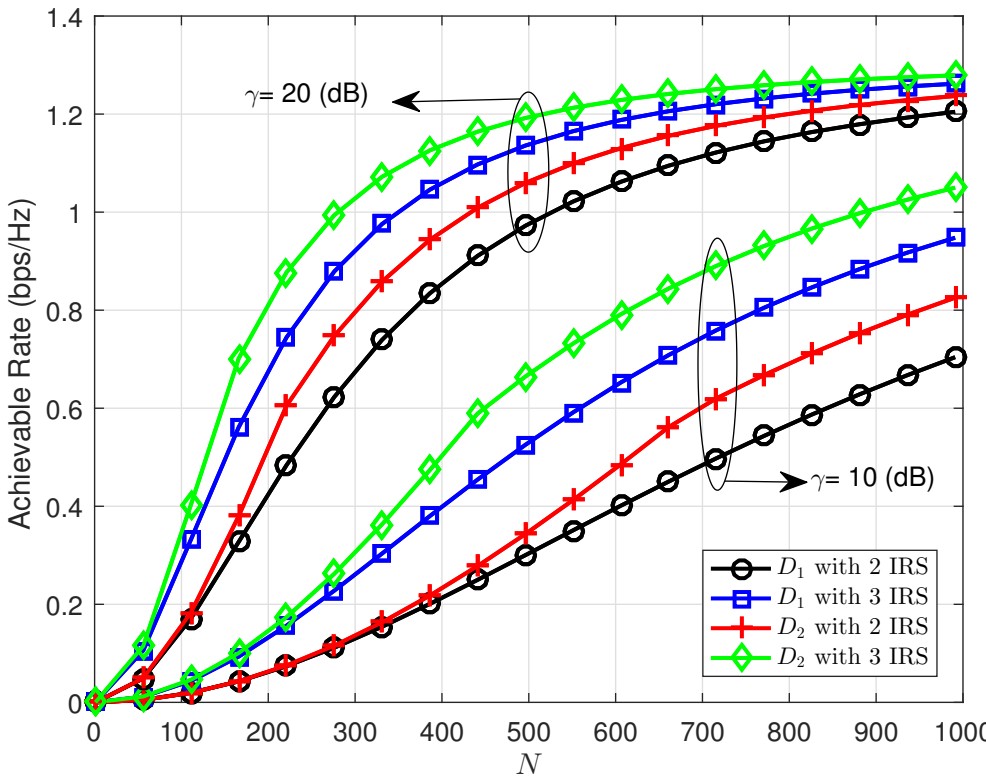

**Figure 3.** The performance of overall achievable rates by varying the number of IRSs ($N = N_1 = N_2 = N_3$) when changing $\gamma$ with Rayleigh fading.

In Figure 4, we confirm the impact of the power allocation factors on the achievable rate. It can be easily seen that more power $\chi_1$ assigned to the first user results in a higher achievable rate while the performance of the second user increases at $\chi_1 = 0.65$ for the case of $\gamma = 20$ (dB) and declines remarkably afterward. The number of meta-surfaces in the three-IRSs case plays an important role in enhancing the quality of received signals at destinations. However, more power assigned to the transmit signal to a selected user is a key factor affecting the achievable rate. On the other hand, we can observe that more IRSs served for user $D_1$ can be employed to improve the rate performance significantly if $\chi_1 = 0.65$ is greater than 0.7.

Furthermore, Figure 5 showcases the severity of the hardware impairment for ergodic performance. In particular, due to the transceiver hardware impairment $\Upsilon_S = \Upsilon_{D_1} = \Upsilon_{D_2}$, the ergodic performance saturates for the case of $\Upsilon_S = \Upsilon_{D_1} = \Upsilon_{D_2} = 0.5$ and cannot be further improved even when increasing $\gamma$ over 30 (dB). It is observed at the low region of $\gamma$ that different levels of transceiver hardware impairment lead to a small gap of achievable rates, and the performance gap among two users is small as well. The capacity ceiling of both users exists when $\Upsilon_S = \Upsilon_{D_1} = \Upsilon_{D_2} = 0.5$ corresponding to $\gamma = 30$ (dB). It is strongly confirmed that the capacity ceiling is determined by the levels of transceiver hardware impairment rather than $N$, $\gamma$ and other scenarios.

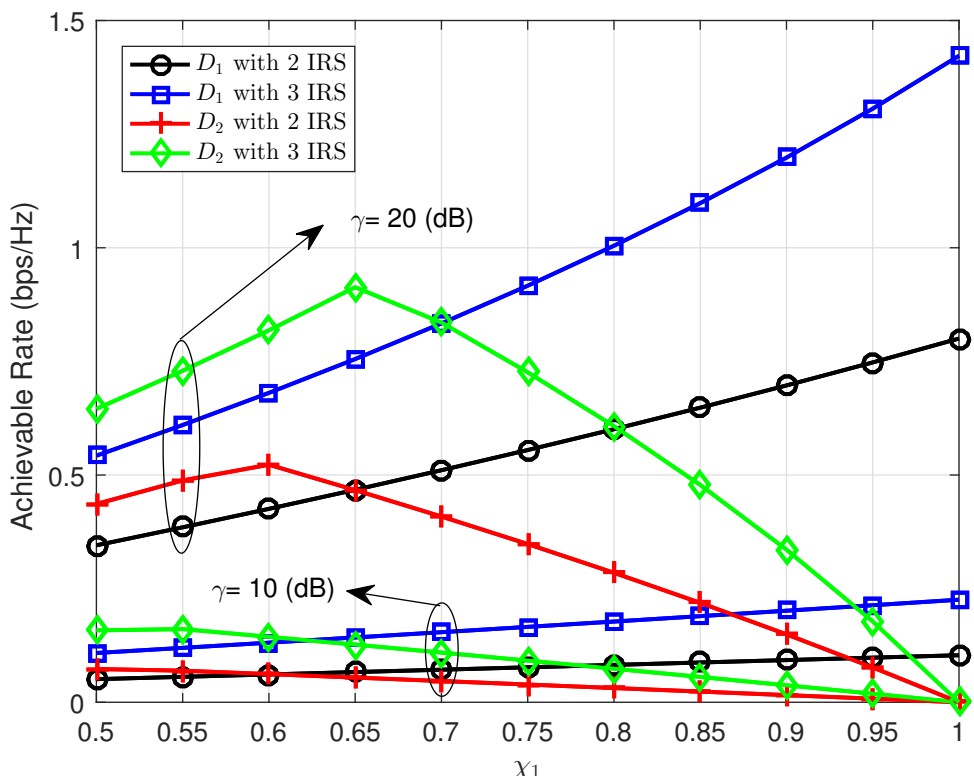

**Figure 4.** Impact of power allocation factors $\chi_1$ on overall achievable rates by varying $\gamma$ with Rayleigh fading.

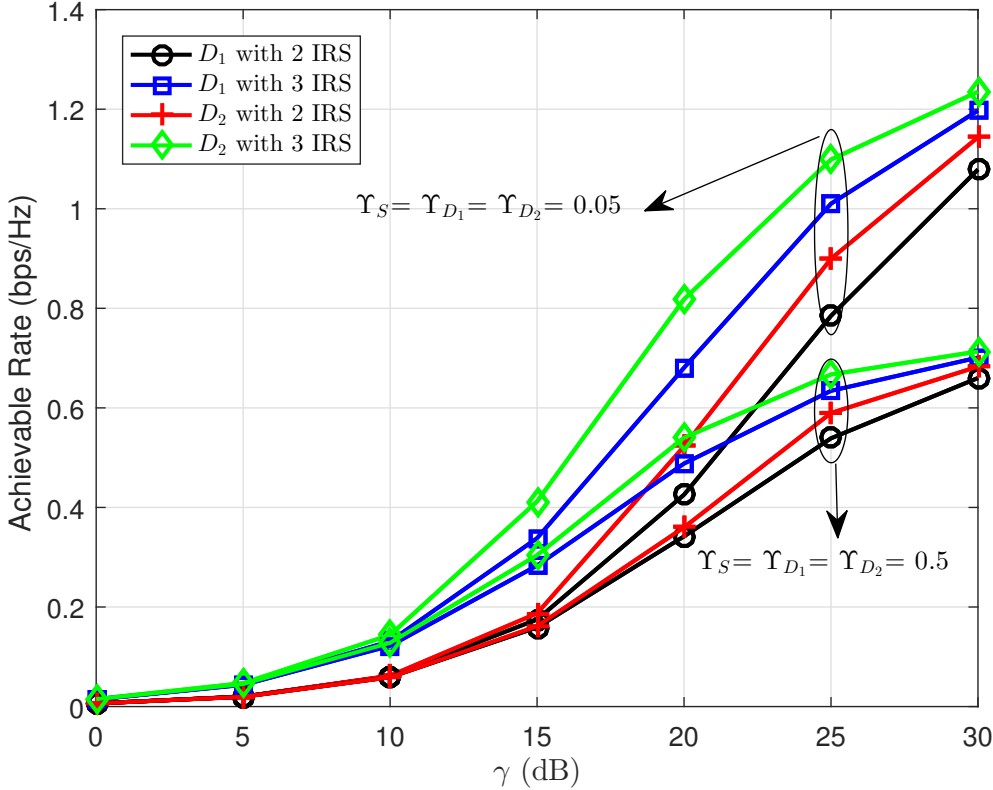

**Figure 5.** Impact of hardware impairments $\Upsilon_S = \Upsilon_{D_1} = \Upsilon_{D_2}$ on overall achievable rates by varying $\gamma$ with Rayleigh fading.

Figure 6 considers the distances of the source–IRS and IRS–destination links. If IRS is placed at the middle point between the source and destinations, the curves of the achievable rate exhibit the lowest values. When the IRS moves close to either the source or the destinations, the achievable rate improves significantly for the case of $\alpha = 2.5$. In this experiment, the achievable rate of the three-IRSs case is still better than the other case.

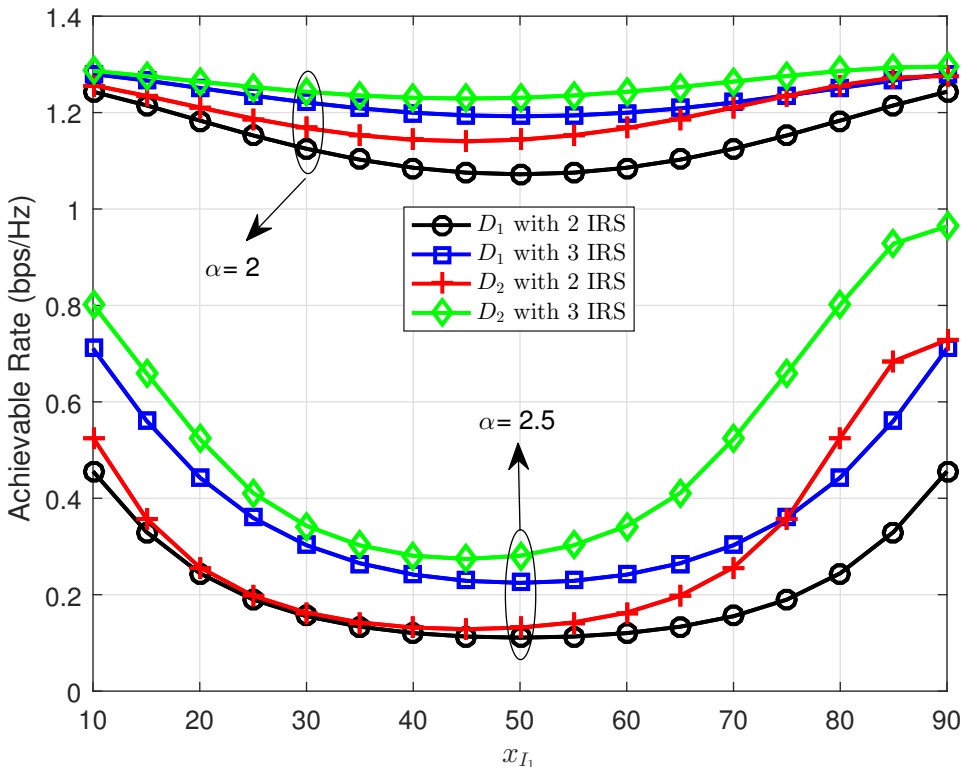

**Figure 6.** Impact on the performance of overall achievable rates by varying $x_{I_1} = x_{I_2} = x_{I_3}$ when changing $\alpha$ with Rayleigh fading.

We can verify how the setting of IRS affects the achievable rate, as shown in Figure 7. At the value of the number of meta-surface $N = 400$, the achievable rate obtains the maximum value. The gap among two cases of $\chi_1$ could be larger when $N$ is greater than 150. The important conclusion is that it is not necessary to design too many meta-surfaces at the IRS since the achievable rate saturates in the region where $N$ is greater than 500.

The setting of the channel has slight impacts on the ergodic performance since we see a small gap among the curves in Figure 8. Besides the locations of IRS, the quality of the channel gives slight variations in the performance of IoT users. Our similar result in the double-IRS and three-IRSs cases can be compared with the results in the recent work [30]. This means that low-cost IoT can obtain benefits with the double-IRS approach with little influence from selecting the channel models.

The setting of IRS, i.e., the reflection coefficient, has the main effect on the achievable rates for two cases of IRS, as shown in Figure 9. We can see that $\eta = 0.8$ corresponds to a higher achievable rate. The reason is that reflecting the capability of IRS could be better for the case of $\eta = 0.8$, and SNDR can be enhanced along with a corresponding improvement of the achievable rate. This simulation still confirms that the three-IRSs case provides higher ergodic performance. However, increasing IRSs leads to a higher cost of deployment. Similarly, Figure 10 confirms that there is a little influence of Rayleigh and Rician channels on ergodic performance for both IoT users.

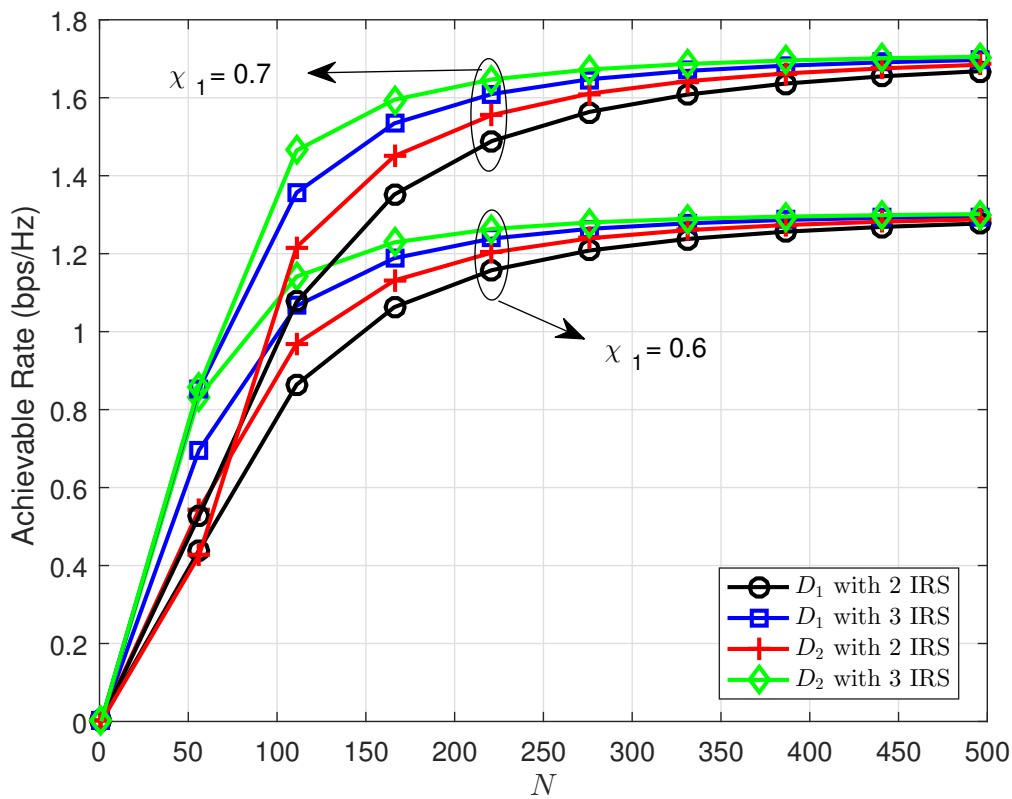

**Figure 7.** The achievable rates versus $N = N_1 = N_2 = N_3$ when changing $\chi_1$ with Rician fading.

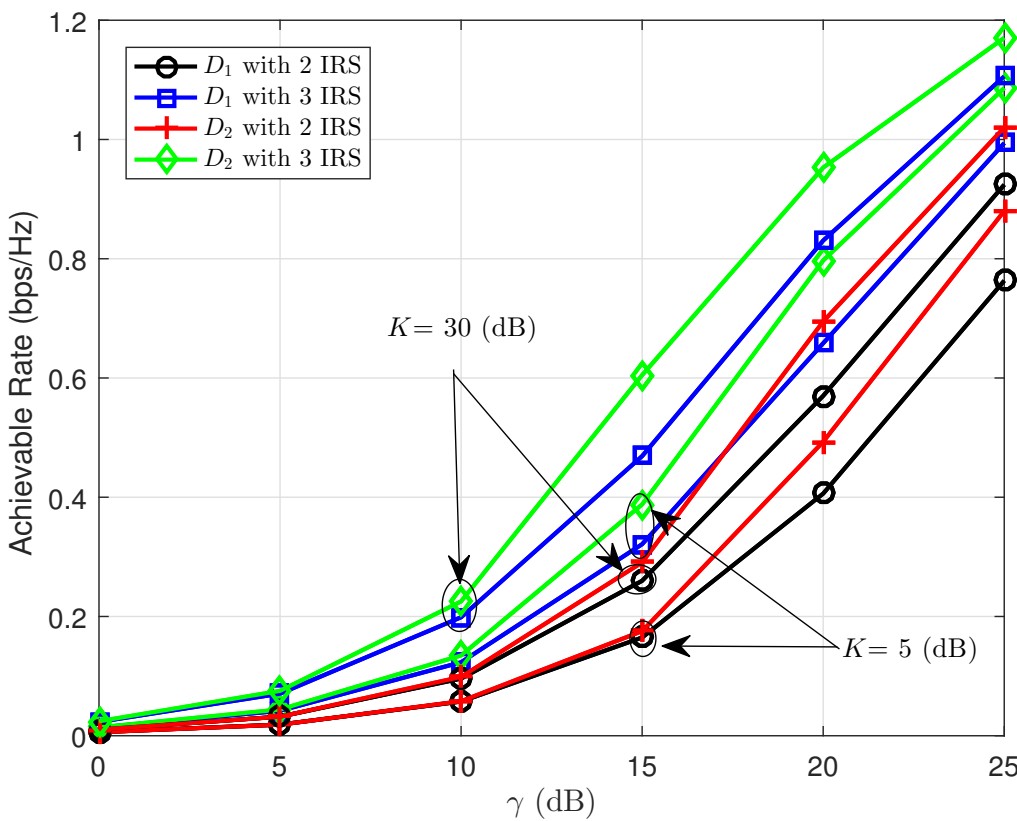

**Figure 8.** Impact on the performance of overall achievable rates by varying $\gamma$ when changing the Rician-$K$ factor.

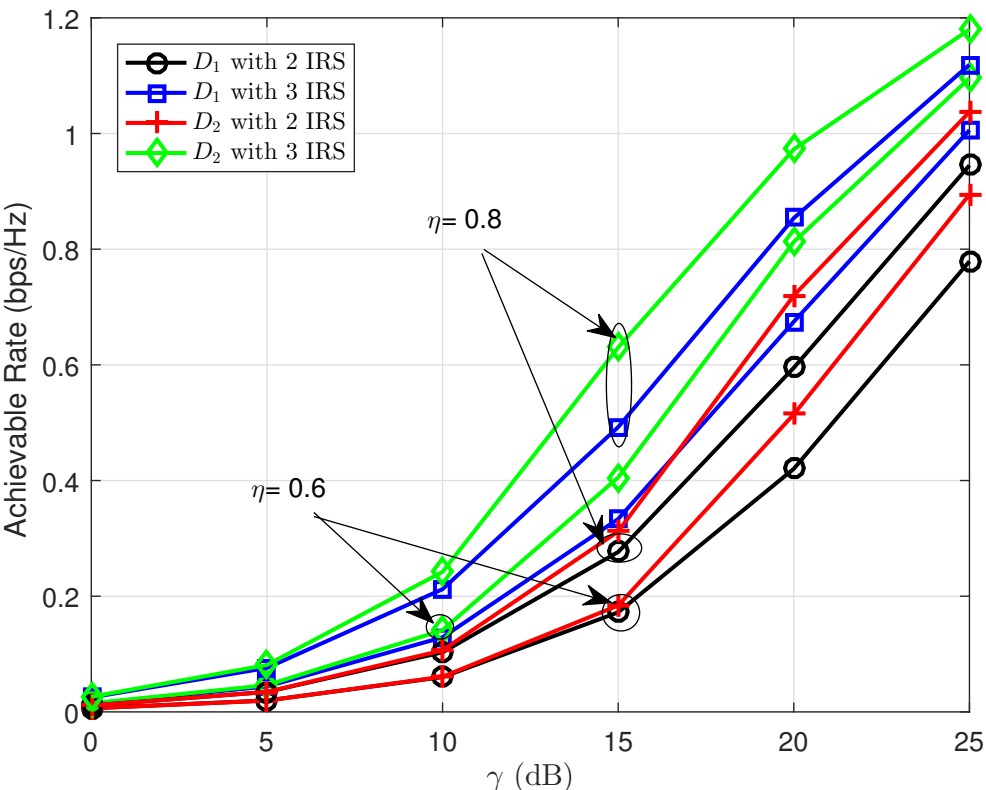

**Figure 9.** Impact on the performance of overall achievable rates by varying $\gamma$ when changing $\eta$ with Rician fading.

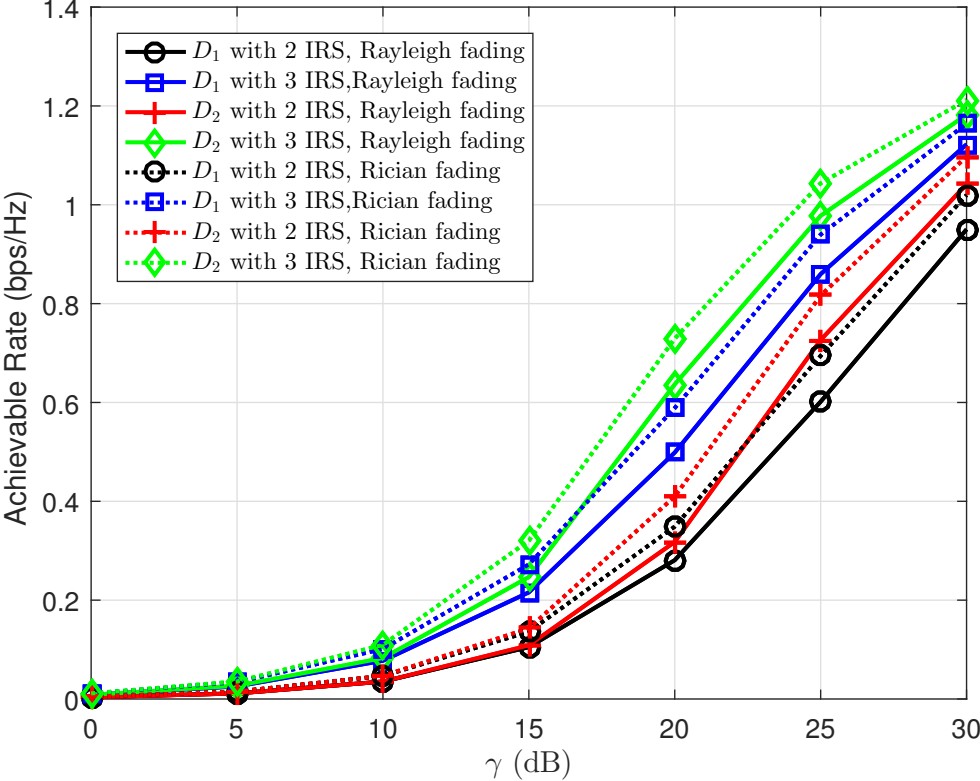

**Figure 10.** Impact on the performance of overall achievable rates of Rayleigh fading and Rician fading by varying $\gamma$.

The benefit of NOMA regarding spectrum efficiency leads to improved performance in terms of the achievable rate, as shown in Figure 11. When the region of $\gamma$ is higher than 20 (dB), a bigger gap between NOMA–IRS and OMA–IRS can be reported. The cases of Rayleigh fading and Rician fading do not affect how much OMA–IRS outperforms OMA–IRS.

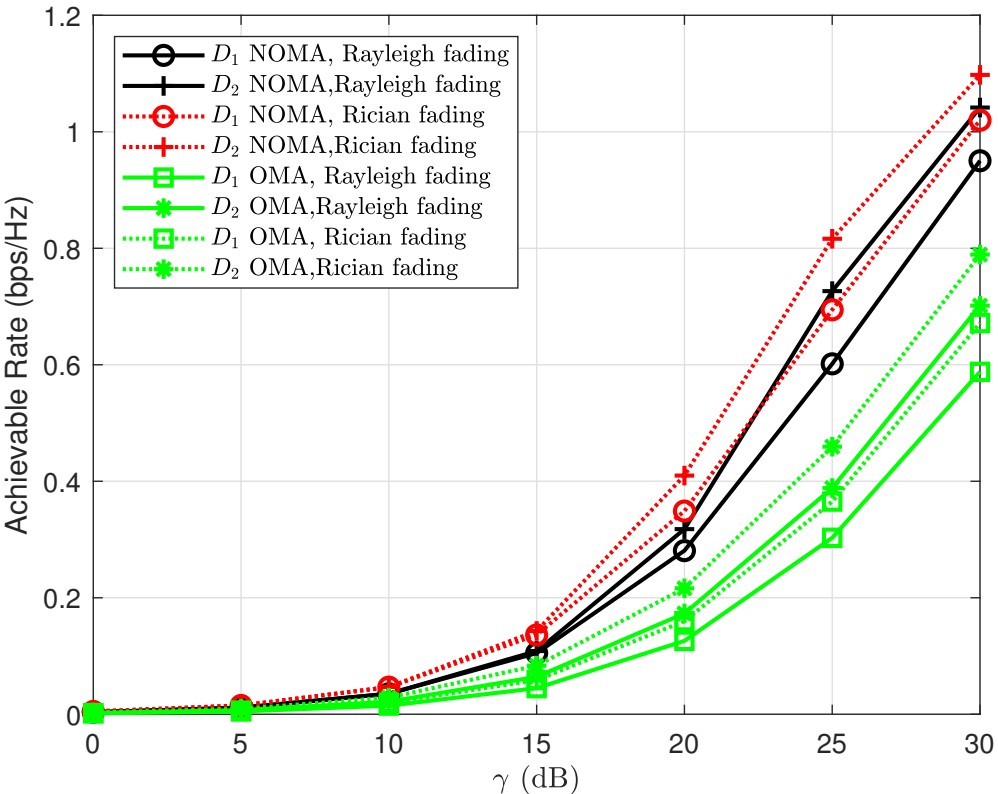

**Figure 11.** Impact on the performance of overall achievable rates of Rayleigh fading and Rician fading between NOMA and OMA with two IRSs by varying $\gamma$.

## 5. Conclusions

In this paper, we have recommended the deployment of a double-IRS NOMA–UAV for IoT downlink, since this approach is sufficient to improve performance at destinations. The achievable rates for many scenarios are evaluated carefully to indicate the main impacts under the limitation of imperfect hardware. In contrast to the existing contributions on NOMA transmission design, this paper focused on analytical analysis by characterizing ergodic performance under several channel distributions. The approximate expressions of the achievable rate were verified precisely since these results are similar to those reported in recent work. We provided extensive numerical results of achievable rates with varying hardware impairment levels, power allocation coefficients and channel parameters to validate the accuracy of our derived results. Better ergodic performance can be obtained when the locations of IRSs are close to the source or IoT device. We confirmed that the crucial role of the UAV is to extend the coverage of the base station. We can deploy UAVs in smart city applications since UAVs can be easily deployed in cases of dense devices, buildings and vehicles in cities. In future work, we plan to deal with multiple antennas equipped at the source with respect to improving the reliable transmission of IoT IRS–NOMA–UAV systems.

**Author Contributions:** M.-S.V.N. contributed to developing the mathematical analysis and performed the simulation experiments; D.-T.D. introduced the idea and contributed to developing the mathematical analysis; V.-D.P. contributed to preparing the manuscript and provided valuable comments; W.U.K. introduced the idea and provided valuable comments; A.L.I. introduced the idea

and provided valuable comments; M.M.F. introduced the idea and provided valuable comments. All authors have read and agreed to the published version of the manuscript.

**Funding:** Van Lang University, Vietnam provided the budget for this study.

**Institutional Review Board Statement:** Not applicable.

**Informed Consent Statement:** Not applicable.

**Data Availability Statement:** Not applicable.

**Acknowledgments:** We are greatly thankful to Van Lang University, Vietnam for providing the budget for this study. The work of Agbotiname Lucky Imoize is supported in part by the Nigerian Petroleum Technology Development Fund (PTDF) and in part by the German Academic Exchange Service (DAAD) through the Nigerian-German Postgraduate Program under grant 57473408.

**Conflicts of Interest:** The authors declare no conflict of interest.

## Appendix A

**Proof.** Now ,defining $\sqrt{\widetilde{A_1^{(1)}}} = \frac{\eta}{N_1} d_{SI_1}^{-\alpha/2} d_{I_1 D_1}^{-\alpha/2} \sum_{n=1}^{N_1} \bar{h}_{SI_1,n} \bar{h}_{I_1 D_1,n}$, and $\sqrt{\widetilde{A_2^{(1)}}} = \frac{\eta}{N_2} d_{SI_2}^{-\alpha/2} d_{I_2 D_1}^{-\alpha/2}$ $\sum_{n=1}^{N_2} \bar{h}_{SI_2,n} \bar{h}_{I_2 D_1,n}$ [36]. Both $\widetilde{A_1^{(1)}}$ and $\widetilde{A_2^{(1)}}$ follow a non-central chi-square distribution with mean values given as [36]

$$\mathrm{E}\left[\widetilde{A_1^{(1)}}\right] = \frac{\eta^2 \left[\pi^2 + (1/N_1)(16 - \pi^2)\right]}{16 d_{SI_1}^{\alpha} d_{I_1 D_1}^{\alpha}}, \tag{A1}$$

and

$$\mathrm{E}\left[\widetilde{A_2^{(1)}}\right] = \frac{\eta^2 \left[\pi^2 + (1/N_2)(16 - \pi^2)\right]}{16 d_{SI_2}^{\alpha} d_{I_2 D_1}^{\alpha}}. \tag{A2}$$

From (5), (A1) and (A2), the expected value of $\gamma_{D_1}^{(1)}$ can be derived as

$$
\begin{aligned}
\mathrm{E}\left[\widetilde{\gamma_{D_1}^{(1)}}\right] &= \mathrm{E}\left[\frac{\gamma\chi_1 \left(N_1 \sqrt{\widetilde{A_1^{(1)}}} + N_2 \sqrt{\widetilde{A_2^{(1)}}}\right)^2}{\gamma\chi_2 \left(N_1 \sqrt{\widetilde{A_1^{(1)}}} + N_2 \sqrt{\widetilde{A_2^{(1)}}}\right)^2 + \left(Y_S^2 + Y_{D_1}^2\right)\gamma\left(N_1 \sqrt{\widetilde{A_1^{(1)}}} + N_2 \sqrt{\widetilde{A_2^{(1)}}}\right)^2 + 1}\right] \\
&= \frac{\gamma\chi_1 \left(N_1^2 \mathrm{E}\left[\widetilde{A_1^{(1)}}\right] + 2N_1 N_2 \mathrm{E}\left[\sqrt{\widetilde{A_1^{(1)}}} \sqrt{\widetilde{A_2^{(1)}}}\right] + N_2^2 \mathrm{E}\left[\widetilde{A_2^{(1)}}\right]\right)}{\gamma\chi_2 \left(N_1^2 \mathrm{E}\left[\widetilde{A_1^{(1)}}\right] + 2N_1 N_2 \mathrm{E}\left[\sqrt{\widetilde{A_1^{(1)}}} \sqrt{\widetilde{A_2^{(1)}}}\right] + N_2^2 \mathrm{E}\left[\widetilde{A_2^{(1)}}\right]\right) + \left(Y_S^2 + Y_{D_1}^2\right)\gamma\left(N_1^2 \mathrm{E}\left[\widetilde{A_1^{(1)}}\right] + 2N_1 N_2 \mathrm{E}\left[\sqrt{\widetilde{A_1^{(1)}}} \sqrt{\widetilde{A_2^{(1)}}}\right] + N_2^2 \mathrm{E}\left[\widetilde{A_2^{(1)}}\right]\right) + 1}.
\end{aligned} \tag{A3}
$$

We let $\Xi_1 = N_1^2 \mathrm{E}\left[\widetilde{A_1^{(1)}}\right] + 2N_1 N_2 \mathrm{E}\left[\sqrt{\widetilde{A_1^{(1)}}} \sqrt{\widetilde{A_2^{(1)}}}\right] + N_2^2 \mathrm{E}\left[\widetilde{A_2^{(1)}}\right]$. From (A1) and (A2), $\Xi_1$ can given as

$$
\begin{aligned}
\Xi_1 &= N_1^2 \frac{\eta^2 \left[\pi^2 + (1/N_1)(16 - \pi^2)\right]}{16 d_{SI_1}^{\alpha} d_{I_1 D_1}^{\alpha}} + 2N_1 N_2 \sqrt{\frac{\eta^2 \left[\pi^2 + (1/N_1)(16 - \pi^2)\right]}{16 d_{SI_1}^{\alpha} d_{I_1 D_1}^{\alpha}}} \sqrt{\frac{\eta^2 \left[\pi^2 + (1/N_2)(16 - \pi^2)\right]}{16 d_{SI_2}^{\alpha} d_{I_2 D_1}^{\alpha}}} + N_2^2 \frac{\eta^2 \left[\pi^2 + (1/N_2)(16 - \pi^2)\right]}{16 d_{SI_2}^{\alpha} d_{I_2 D_1}^{\alpha}} \\
&= \frac{1}{16} d_{SI_1}^{-\alpha} d_{I_1 D_1}^{-\alpha} \eta^2 N_1 \left[N_1 \pi^2 + 16 - \pi^2\right] + \frac{1}{8} d_{SI_1}^{-\alpha/2} d_{I_1 D_1}^{-\alpha/2} d_{SI_2}^{-\alpha/2} d_{I_2 D_1}^{-\alpha/2} \eta\eta \sqrt{N_1 \left[N_1 \pi^2 + 16 - \pi^2\right]} \\
&\quad \times \sqrt{N_2 \left[N_2 \pi^2 + 16 - \pi^2\right]} + \frac{1}{16} d_{SI_2}^{-\alpha} d_{I_2 D_1}^{-\alpha} \eta^2 N_2 \left[N_2 \pi^2 + 16 - \pi^2\right] \\
&= \frac{1}{16} d_{SI_1}^{-\alpha} d_{I_1 D_1}^{-\alpha} \eta^2 N_1 \left[16 + (N_1 - 1)\pi^2\right] + \frac{1}{8} d_{SI_1}^{-\alpha/2} d_{I_1 D_1}^{-\alpha/2} d_{SI_2}^{-\alpha/2} d_{I_2 D_1}^{-\alpha/2} \eta\eta \\
&\quad \times \sqrt{N_1 N_2 \left[16 + (N_1 - \pi^2)\pi^2\right] \left[16 + (N_2 - \pi^2)\pi^2\right]} + \frac{1}{16} d_{SI_2}^{-\alpha} d_{I_2 D_1}^{-\alpha} \eta^2 N_2 \left[16 + (N_2 - \pi^2)\pi^2\right] \\
&= \frac{1}{16}\beta_1 + \frac{1}{8}\beta_2 + \frac{1}{16}\beta_3,
\end{aligned} \tag{A4}
$$

where $\beta_1 = d_{SI_1}^{-\alpha} d_{I_1 D_1}^{-\alpha} \eta^2 N_1 \left(16 + (N_1 - 1)\pi^2\right)$, $\beta_2 = d_{SI_1}^{-\alpha/2} d_{I_1 D_1}^{-\alpha/2} d_{SI_2}^{-\alpha/2} d_{I_2 D_1}^{-\alpha/2} \eta\eta$ $\sqrt{N_1 N_2 (16 + (N_1 - 1)\pi^2)(16 + (N_2 - 1)\pi^2)}$, $\beta_3 = d_{SI_2}^{-\alpha} d_{I_2 D_1}^{-\alpha} \eta^2 N_2 \left(16 + (N_2 - 1)\pi^2\right)$.

From (A4) into (A3), $\mathrm{E}\left[\widetilde{\gamma_{D_1}^{(1)}}\right]$ is formulated by

$$
\mathrm{E}\left[\widetilde{\gamma_{D_1}^{(1)}}\right] \leq \frac{\gamma\chi_1\left(\frac{1}{16}\beta_1+\frac{1}{8}\beta_2+\frac{1}{16}\beta_3\right)}{\gamma\chi_2\left(\frac{1}{16}\beta_1+\frac{1}{8}\beta_2+\frac{1}{16}\beta_3\right)+\left(\mathrm{Y}_S^2+\mathrm{Y}_{D_1}^2\right)\gamma\left(\frac{1}{16}\beta_1+\frac{1}{8}\beta_2+\frac{1}{16}\beta_3\right)+1}. \tag{A5}
$$

This is the end of the proof. $\square$

**Appendix B**

**Proof.** Now, defining $\sqrt{\widetilde{B_1^{(1)}}} = \frac{\eta}{N_1}d_{SI_1}^{-\alpha/2}d_{I_1D_2}^{-\alpha/2}\sum\limits_{n=1}^{N_1}\bar{h}_{SI_1,n}\bar{h}_{I_1D_2,n}$, and $\sqrt{\widetilde{B_2^{(1)}}} = \frac{\eta}{N_2}d_{SI_2}^{-\alpha/2}d_{I_2D_2}^{-\alpha/2}$

$\sum\limits_{n=1}^{N_2}\bar{h}_{SI_2,n}\bar{h}_{I_2D_2,n}$ [36]. Both $\widetilde{B_1^{(1)}}$ and $\widetilde{B_2^{(1)}}$ follow a non-central chi-square distribution with mean values given as [36]

$$
\mathrm{E}\left[\widetilde{B_1^{(1)}}\right] = \frac{\eta^2\left[\pi^2+(1/N_1)(16-\pi^2)\right]}{16d_{SI_1}^\alpha d_{I_1D_2}^\alpha}, \tag{A6}
$$

and

$$
\mathrm{E}\left[\widetilde{B_2^{(1)}}\right] = \frac{\eta^2\left[\pi^2+(1/N_2)(16-\pi^2)\right]}{16d_{SI_2}^\alpha d_{I_2D_2}^\alpha}. \tag{A7}
$$

From (10), the expected value of $\gamma_{x_1,D_2}^{(1)}$ can be derived as

$$
\begin{aligned}
\mathrm{E}\left[\widetilde{\gamma_{x_1,D_2}^{(1)}}\right] &= \mathrm{E}\left[\frac{\gamma\chi_1\left(N_1\sqrt{\widetilde{B_1^{(1)}}}+N_2\sqrt{\widetilde{B_2^{(1)}}}\right)^2}{\gamma\chi_2\left(N_1\sqrt{\widetilde{B_1^{(1)}}}+N_2\sqrt{\widetilde{B_2^{(1)}}}\right)^2+\left(\mathrm{Y}_S^2+\mathrm{Y}_{D_2}^2\right)\gamma\left(N_1\sqrt{\widetilde{B_1^{(1)}}}+N_2\sqrt{\widetilde{B_2^{(1)}}}\right)^2+1}\right] \\
&= \frac{\gamma\chi_1\left(N_1^2\mathrm{E}\left[\widetilde{B_1^{(1)}}\right]+2N_1N_2\mathrm{E}\left[\sqrt{\widetilde{B_1^{(1)}}}\sqrt{\widetilde{B_2^{(1)}}}\right]+N_2^2\mathrm{E}\left[\widetilde{B_2^{(1)}}\right]\right)}{\gamma\chi_2\left(N_1^2\mathrm{E}\left[\widetilde{B_1^{(1)}}\right]+2N_1N_2\mathrm{E}\left[\sqrt{\widetilde{B_1^{(1)}}}\sqrt{\widetilde{B_2^{(1)}}}\right]+N_2^2\mathrm{E}\left[\widetilde{B_2^{(1)}}\right]\right)+\left(\mathrm{Y}_S^2+\mathrm{Y}_{D_2}^2\right)\gamma\left(N_1^2\mathrm{E}\left[\widetilde{B_1^{(1)}}\right]+2N_1N_2\mathrm{E}\left[\sqrt{\widetilde{B_1^{(1)}}}\sqrt{\widetilde{B_2^{(1)}}}\right]+N_2^2\mathrm{E}\left[\widetilde{B_2^{(1)}}\right]\right)+1}.
\end{aligned} \tag{A8}
$$

We let $\Xi_2 = N_1^2\mathrm{E}\left[\widetilde{B_1^{(1)}}\right]+2N_1N_2\mathrm{E}\left[\sqrt{\widetilde{B_1^{(1)}}}\sqrt{\widetilde{B_2^{(1)}}}\right]+N_2^2\mathrm{E}\left[\widetilde{B_2^{(1)}}\right]$. From (A6) and (A7), $\Xi_1$ can given as

$$
\begin{aligned}
\Xi_2 &= N_1^2\frac{\eta^2\left[\pi^2+(1/N_1)(16-\pi^2)\right]}{16d_{SI_1}^\alpha d_{I_1D_2}^\alpha}+2N_1N_2\sqrt{\frac{\eta^2\left[\pi^2+(1/N_1)(16-\pi^2)\right]}{16d_{SI_1}^\alpha d_{I_1D_2}^\alpha}}\sqrt{\frac{\eta^2\left[\pi^2+(1/N_2)(16-\pi^2)\right]}{16d_{SI_2}^\alpha d_{I_2D_2}^\alpha}}+N_2^2\frac{\eta^2\left[\pi^2+(1/N_2)(16-\pi^2)\right]}{16d_{SI_2}^\alpha d_{I_2D_2}^\alpha} \\
&= \frac{1}{16}d_{SI_1}^{-\alpha}d_{I_1D_2}^{-\alpha}\eta^2N_1\left[N_1\pi^2+16-\pi^2\right]+\frac{1}{8}d_{SI_1}^{-\alpha/2}d_{I_1D_2}^{-\alpha/2}d_{SI_2}^{-\alpha/2}d_{I_2D_2}^{-\alpha/2}\eta\eta N_1N_2 \\
&\quad\times\sqrt{N_1\left[N_1\pi^2+16-\pi^2\right]}\sqrt{N_2\left[N_2\pi^2+16-\pi^2\right]}+\frac{1}{16}d_{SI_2}^{-\alpha}d_{I_2D_2}^{-\alpha}\eta^2N_2\left[N_2\pi^2+16-\pi^2\right] \\
&= \frac{1}{16}d_{SI_1}^{-\alpha}d_{I_1D_2}^{-\alpha}\eta^2N_1\left[16+(N_1-1)\pi^2\right]+\frac{1}{8}d_{SI_1}^{-\alpha/2}d_{I_1D_2}^{-\alpha/2}d_{SI_2}^{-\alpha/2}d_{I_2D_2}^{-\alpha/2}\eta\eta N_1N_2 \\
&\quad\times\sqrt{N_1N_2[16+(N_1-1)\pi^2][16+(N_2-1)\pi^2]}+\frac{1}{16}d_{SI_2}^{-\alpha}d_{I_2D_2}^{-\alpha}\eta^2N_2\left[16+(N_2-1)\pi^2\right] \\
&= \frac{1}{16}\psi_1+\frac{1}{8}\psi_2+\frac{1}{16}\psi_3,
\end{aligned} \tag{A9}
$$

in which $\psi_1 = d_{SI_1}^{-\alpha}d_{I_1D_2}^{-\alpha}\eta^2N_1\left(16+(N_1-1)\pi^2\right)$, $\psi_2 = d_{SI_1}^{-\alpha/2}d_{I_1D_2}^{-\alpha/2}d_{SI_2}^{-\alpha/2}d_{I_2D_2}^{-\alpha/2}\eta\eta$ $\sqrt{N_1N_2(16+(N_1-1)\pi^2)(16+(N_2-1)\pi^2)}$, $\psi_3 = d_{SI_2}^{-\alpha}d_{I_2D_2}^{-\alpha}\eta^2N_2\left(16+(N_2-1)\pi^2\right)$.

From (A9) and (A8), $\mathrm{E}\left[\widetilde{\gamma_{x_1,D_2}^{(1)}}\right]$ is formulated by

$$
\mathrm{E}\left[\widetilde{\gamma_{x_1,D_2}^{(1)}}\right] \leq \frac{\gamma\chi_1\left(\frac{1}{16}\psi_1+\frac{1}{8}\psi_2+\frac{1}{16}\psi_3\right)}{\gamma\chi_2\left(\frac{1}{16}\psi_1+\frac{1}{8}\psi_2+\frac{1}{16}\psi_3\right)+\left(\mathrm{Y}_S^2+\mathrm{Y}_{D_2}^2\right)\gamma\left(\frac{1}{16}\psi_1+\frac{1}{8}\psi_2+\frac{1}{16}\psi_3\right)+1}. \tag{A10}
$$

From (11) and similarly (A8), the expected value of $\gamma_{x_2,D_2}^{(1)}$ can be derived as

$$
\begin{aligned}
\mathrm{E}\left[\widetilde{\gamma_{x_2,D_2}^{(1)}}\right] &= \mathrm{E}\left[\frac{\gamma\chi_2\left(N_1\sqrt{\widetilde{B_1^{(1)}}}+N_2\sqrt{\widetilde{B_2^{(1)}}}\right)^2}{\left(\mathrm{Y}_S^2+\mathrm{Y}_{D_2}^2\right)\gamma\left(N_1\sqrt{\widetilde{B_1^{(1)}}}+N_2\sqrt{\widetilde{B_2^{(1)}}}\right)^2+1}\right] \\
&= \frac{\gamma\chi_2\left(N_1^2\mathrm{E}\left[\widetilde{B_1^{(1)}}\right]+2N_1N_2\mathrm{E}\left[\sqrt{\widetilde{B_1^{(1)}}}\sqrt{\widetilde{B_2^{(1)}}}\right]+N_2^2\mathrm{E}\left[\widetilde{B_2^{(1)}}\right]\right)}{\left(\mathrm{Y}_S^2+\mathrm{Y}_{D_2}^2\right)\gamma\left(N_1^2\mathrm{E}\left[\widetilde{B_1^{(1)}}\right]+2N_1N_2\mathrm{E}\left[\sqrt{\widetilde{B_1^{(1)}}}\sqrt{\widetilde{B_2^{(1)}}}\right]+N_2^2\mathrm{E}\left[\widetilde{B_2^{(1)}}\right]\right)+1} \\
&\leq \frac{\gamma\chi_2\left(\frac{1}{16}\psi_1+\frac{1}{8}\psi_2+\frac{1}{16}\psi_3\right)}{\left(\mathrm{Y}_S^2+\mathrm{Y}_{D_2}^2\right)\gamma\left(\frac{1}{16}\psi_1+\frac{1}{8}\psi_2+\frac{1}{16}\psi_3\right)+1}.
\end{aligned}
\tag{A11}
$$

From (A10) and (A11), we can obtain (20).

The proof is completed.　□

**Appendix C**

**Proof.** Now, defining $\sqrt{\widehat{A_1^{(1)}}}=\frac{\eta}{N_1}\rho_{SI_1D_1}d_{SI_1}^{-\alpha/2}d_{I_1D_1}^{-\alpha/2}\sum\limits_{n=1}^{N_1}\widehat{h}_{SI_1,n}\widehat{h}_{I_1D_1,n}$ and $\sqrt{\widehat{A_2^{(1)}}}=\frac{\eta}{N_2}\rho_{SI_2D_1}$

$d_{SI_2}^{-\alpha/2}d_{I_2D_1}^{-\alpha/2}\sum\limits_{n=1}^{N_2}\widehat{h}_{SI_2,n}\widehat{h}_{I_2D_1,n}$ [36], where $\rho_{SI_1D_1}=\sqrt{\frac{K_{SI_1}K_{I_1D_1}}{(K_{SI_1}+1)(K_{I_1D_1}+1)}}$,

$\rho_{SI_2D_1}=\sqrt{\frac{K_{SI_2}K_{I_2D_1}}{(K_{SI_2}+1)(K_{I_2D_1}+1)}}$. Moreover, $\widehat{A_1^{(1)}}$ and $\widehat{A_2^{(1)}}$ have constant mean values as [36]

$$
\mathrm{E}\left[\widehat{A_1^{(1)}}\right]=\frac{\eta^2\rho_{SI_1D_1}^2}{d_{SI_1}^\alpha d_{I_1D_1}^\alpha},
\tag{A12}
$$

$$
\mathrm{E}\left[\widehat{A_2^{(1)}}\right]=\frac{\eta^2\rho_{SI_2D_1}^2}{d_{SI_2}^\alpha d_{I_2D_1}^\alpha}.
\tag{A13}
$$

Therefore, the expected value of $\gamma_{D_1}^{(1)}$ can be derived as

$$
\begin{aligned}
\mathrm{E}\left[\widehat{\gamma_{D_1}^{(1)}}\right] &= \mathrm{E}\left[\frac{\gamma\chi_1\left(N_1\sqrt{\widehat{A_1^{(1)}}}+N_2\sqrt{\widehat{A_2^{(1)}}}\right)^2}{\gamma\chi_2\left(N_1\sqrt{\widehat{A_1^{(1)}}}+N_2\sqrt{\widehat{A_2^{(1)}}}\right)^2+\left(\mathrm{Y}_S^2+\mathrm{Y}_{D_1}^2\right)\gamma\left(N_1\sqrt{\widehat{A_1^{(1)}}}+N_2\sqrt{\widehat{A_2^{(1)}}}\right)^2+1}\right] \\
&= \frac{\gamma\chi_1\left(N_1^2\mathrm{E}\left[\widehat{A_1^{(1)}}\right]+2N_1N_2\mathrm{E}\left[\sqrt{\widehat{A_1^{(1)}}}\sqrt{\widehat{A_2^{(1)}}}\right]+N_2^2\mathrm{E}\left[\widehat{A_2^{(1)}}\right]\right)}{\gamma\chi_2\left(N_1^2\mathrm{E}\left[\widehat{A_1^{(1)}}\right]+2N_1N_2\mathrm{E}\left[\sqrt{\widehat{A_1^{(1)}}}\sqrt{\widehat{A_2^{(1)}}}\right]+N_2^2\mathrm{E}\left[\widehat{A_2^{(1)}}\right]\right)+\left(\mathrm{Y}_S^2+\mathrm{Y}_{D_1}^2\right)\gamma\left(N_1^2\mathrm{E}\left[\widehat{A_1^{(1)}}\right]+2N_1N_2\mathrm{E}\left[\sqrt{\widehat{A_1^{(1)}}}\sqrt{\widehat{A_2^{(1)}}}\right]+N_2^2\mathrm{E}\left[\widehat{A_2^{(1)}}\right]\right)+1}.
\end{aligned}
\tag{A14}
$$

We let $\Xi_3=N_1^2\mathrm{E}\left[\widehat{A_1^{(1)}}\right]+2N_1N_2\mathrm{E}\left[\sqrt{\widehat{A_1^{(1)}}}\sqrt{\widehat{A_2^{(1)}}}\right]+N_2^2\mathrm{E}\left[\widehat{A_2^{(1)}}\right]$. From (A12) and

(A13), $\Xi_3$ can be calculated as

$$
\begin{aligned}
\Xi_3 &= N_1^2\frac{\eta^2\rho_{SI_1D_1}^2}{d_{SI_1}^\alpha d_{I_1D_1}^\alpha}+2N_1N_2\sqrt{\frac{\eta^2\rho_{SI_1D_1}^2}{d_{SI_1}^\alpha d_{I_1D_1}^\alpha}}\sqrt{\frac{\eta^2\rho_{SI_2D_1}^2}{d_{SI_2}^\alpha d_{I_2D_1}^\alpha}}+N_2^2\frac{\eta^2\rho_{SI_2D_1}^2}{d_{SI_2}^\alpha d_{I_2D_1}^\alpha} \\
&= \frac{\eta^2N_1^2\rho_{SI_1D_1}^2}{d_{SI_1}^\alpha d_{I_1D_1}^\alpha}+2\frac{\eta\eta N_1N_2\rho_{SI_1D_1}\rho_{SI_2D_1}}{d_{SI_1}^{\alpha/2}d_{I_1D_1}^{\alpha/2}d_{SI_2}^{\alpha/2}d_{I_2D_1}^{\alpha/2}}+\frac{\eta^2N_2^2\rho_{SI_2D_1}^2}{d_{SI_2}^\alpha d_{I_2D_1}^\alpha} \\
&= \partial_1+2\partial_2+\partial_3,
\end{aligned}
\tag{A15}
$$

where $\partial_1=\frac{\eta^2N_1^2\rho_{SI_1D_1}^2}{d_{SI_1}^\alpha d_{I_1D_1}^\alpha}$, $\partial_2=\frac{\eta\eta N_1N_2\rho_{SI_1D_1}\rho_{SI_2D_1}}{d_{SI_1}^{\alpha/2}d_{I_1D_1}^{\alpha/2}d_{SI_2}^{\alpha/2}d_{I_2D_1}^{\alpha/2}}$, $\partial_3=\frac{\eta^2N_2^2\rho_{SI_2D_1}^2}{d_{SI_2}^\alpha d_{I_2D_1}^\alpha}$.

From (A15) into (A14), $\mathrm{E}\left[\widehat{\gamma_{D_1}^{(1)}}\right]$ is formulated by

$$\mathrm{E}\left[\widehat{\gamma_{D_1}^{(1)}}\right] \leq \frac{\gamma\chi_1(\partial_1+2\partial_2+\partial_3)}{\gamma\chi_2(\partial_1+2\partial_2+\partial_3)+\left(\mathrm{Y}_S^2+\mathrm{Y}_{D_1}^2\right)\gamma(\partial_1+2\partial_2+\partial_3)+1}. \tag{A16}$$

The proof is completed.　□

## Appendix D

**Proof.** Now, defining $\sqrt{\widehat{B_1^{(1)}}} = \frac{\eta}{N_1}\rho_{SI_1D_2}d_{SI_1}^{-\alpha/2}d_{I_1D_2}^{-\alpha/2}\sum\limits_{n=1}^{N_1}\hat{h}_{SI_1,n}\hat{h}_{I_1D_2,n}$ and $\sqrt{\widehat{B_2^{(1)}}} = \frac{\eta}{N_2}\rho_{SI_2D_2}$

$d_{SI_2}^{-\alpha/2}d_{I_2D_2}^{-\alpha/2}\sum\limits_{n=1}^{N_2}\hat{h}_{SI_2,n}\hat{h}_{I_2D_2,n}$　[36],　where　$\rho_{SI_1D_2} = \sqrt{\frac{K_{SI_1}K_{I_1D_2}}{(K_{SI_1}+1)(K_{I_1D_2}+1)}}$,

$\rho_{SI_2D_2} = \sqrt{\frac{K_{SI_2}K_{I_2D_2}}{(K_{SI_2}+1)(K_{I_2D_2}+1)}}$. Moreover, $\widehat{B_1^{(1)}}$ and $\widehat{B_2^{(1)}}$ have constant mean values as [36]

$$\mathrm{E}\left[\widehat{B_1^{(1)}}\right] = \frac{\eta^2\rho_{SI_1D_2}^2}{d_{SI_1}^\alpha d_{I_1D_2}^\alpha}, \tag{A17}$$

$$\mathrm{E}\left[\widehat{B_2^{(1)}}\right] = \frac{\eta^2\rho_{SI_2D_2}^2}{d_{SI_2}^\alpha d_{I_2D_2}^\alpha}. \tag{A18}$$

The expected value of $\gamma_{x_1,D_2}^{(1)}$ can be derived as

$$\mathrm{E}\left[\widehat{\gamma_{x_1,D_2}^{(1)}}\right] = \mathrm{E}\left[\frac{\gamma\chi_1\left(N_1\sqrt{\widehat{B_1^{(1)}}}+N_2\sqrt{\widehat{B_2^{(1)}}}\right)^2}{\gamma\chi_2\left(N_1\sqrt{\widehat{B_1^{(1)}}}+N_2\sqrt{\widehat{B_2^{(1)}}}\right)^2+\left(\mathrm{Y}_S^2+\mathrm{Y}_{D_2}^2\right)\gamma\left(N_1\sqrt{\widehat{B_1^{(1)}}}+N_2\sqrt{\widehat{B_2^{(1)}}}\right)^2+1}\right]$$

$$= \frac{\gamma\chi_1\left(N_1^2\mathrm{E}\left[\widehat{B_1^{(1)}}\right]+2N_1N_2\mathrm{E}\left[\sqrt{\widehat{B_1^{(1)}}}\sqrt{\widehat{B_2^{(1)}}}\right]+N_2^2\mathrm{E}\left[\widehat{B_2^{(1)}}\right]\right)}{\gamma\chi_2\left(N_1^2\mathrm{E}\left[\widehat{B_1^{(1)}}\right]+2N_1N_2\mathrm{E}\left[\sqrt{\widehat{B_1^{(1)}}}\sqrt{\widehat{B_2^{(1)}}}\right]+N_2^2\mathrm{E}\left[\widehat{B_2^{(1)}}\right]\right)+\left(\mathrm{Y}_S^2+\mathrm{Y}_{D_2}^2\right)\gamma\left(N_1^2\mathrm{E}\left[\widehat{B_1^{(1)}}\right]+2N_1N_2\mathrm{E}\left[\sqrt{\widehat{B_1^{(1)}}}\sqrt{\widehat{B_2^{(1)}}}\right]+N_2^2\mathrm{E}\left[\widehat{B_2^{(1)}}\right]\right)+1}. \tag{A19}$$

We let $\Xi_4 = N_1^2\mathrm{E}\left[\widehat{B_1^{(1)}}\right]+2N_1N_2\mathrm{E}\left[\sqrt{\widehat{B_1^{(1)}}}\sqrt{\widehat{B_2^{(1)}}}\right]+N_2^2\mathrm{E}\left[\widehat{B_2^{(1)}}\right]$. From (A17) and (A18),

$\Xi_4$ can be calculated as

$$\begin{aligned}\Xi_4 &= N_1^2\frac{\eta^2\rho_{SI_1D_2}^2}{d_{SI_1}^\alpha d_{I_1D_2}^\alpha}+2N_1N_2\sqrt{\frac{\eta^2\rho_{SI_1D_2}^2}{d_{SI_1}^\alpha d_{I_1D_2}^\alpha}}\sqrt{\frac{\eta^2\rho_{SI_2D_2}^2}{d_{SI_2}^\alpha d_{I_2D_2}^\alpha}}+N_2^2\frac{\eta^2\rho_{SI_2D_2}^2}{d_{SI_2}^\alpha d_{I_2D_2}^\alpha}\\&=\frac{\eta^2N_1^2\rho_{SI_1D_2}^2}{d_{SI_1}^\alpha d_{I_1D_2}^\alpha}+2\frac{\eta\eta N_1N_2\rho_{SI_1D_2}\rho_{SI_2D_2}}{d_{SI_1}^{\alpha/2}d_{I_1D_2}^{\alpha/2}d_{SI_2}^{\alpha/2}d_{I_2D_2}^{\alpha/2}}+\frac{\eta^2N_2^2\rho_{SI_2D_2}^2}{d_{SI_2}^\alpha d_{I_2D_2}^\alpha}\\&=\omega_1+2\omega_2+\omega_3,\end{aligned} \tag{A20}$$

where $\omega_1 = \frac{\eta^2N_1^2\rho_{SI_1D_2}^2}{d_{SI_1}^\alpha d_{I_1D_2}^\alpha}$, $\omega_2 = \frac{\eta\eta N_1N_2\rho_{SI_1D_2}\rho_{SI_2D_2}}{d_{SI_1}^{\alpha/2}d_{I_1D_2}^{\alpha/2}d_{SI_2}^{\alpha/2}d_{I_2D_2}^{\alpha/2}}$, $\omega_3 = \frac{\eta^2N_2^2\rho_{SI_2D_2}^2}{d_{SI_2}^\alpha d_{I_2D_2}^\alpha}$.

From (A20) and (A19), $\mathrm{E}\left[\widehat{\gamma_{x_1,D_2}^{(1)}}\right]$ is formulated by

$$\mathrm{E}\left[\widehat{\gamma_{x_1,D_2}^{(1)}}\right] \leq \frac{\gamma\chi_1(\omega_1+2\omega_2+\omega_3)}{\gamma\chi_2(\omega_1+2\omega_2+\omega_3)+\left(\mathrm{Y}_S^2+\mathrm{Y}_{D_2}^2\right)\gamma(\omega_1+2\omega_2+\omega_3)+1}. \tag{A21}$$

Similar to (A19), the expected value of $\gamma_{x_2,D_2}^{(1)}$ can be derived as

$$
\begin{aligned}
\mathrm{E}\left[\widehat{\gamma_{x_2,D_2}^{(1)}}\right] &= \mathrm{E}\left[\frac{\gamma\chi_2\left(N_1\sqrt{\widehat{B_1^{(1)}}}+N_2\sqrt{\widehat{B_2^{(1)}}}\right)^2}{\left(\mathrm{Y}_S^2+\mathrm{Y}_{D_2}^2\right)\gamma\left(N_1\sqrt{\widehat{B_1^{(1)}}}+N_2\sqrt{\widehat{B_2^{(1)}}}\right)^2+1}\right] \\
&= \frac{\gamma\chi_2\left(N_1^2\mathrm{E}\left[\widehat{B_1^{(1)}}\right]+2N_1N_2\mathrm{E}\left[\sqrt{\widehat{B_1^{(1)}}}\sqrt{\widehat{B_2^{(1)}}}\right]+N_2^2\mathrm{E}\left[\widehat{B_2^{(1)}}\right]\right)}{\left(\mathrm{Y}_S^2+\mathrm{Y}_{D_2}^2\right)\gamma\left(N_1^2\mathrm{E}\left[\widehat{B_1^{(1)}}\right]+2N_1N_2\mathrm{E}\left[\sqrt{\widehat{B_1^{(1)}}}\sqrt{\widehat{B_2^{(1)}}}\right]+N_2^2\mathrm{E}\left[\widehat{B_2^{(1)}}\right]\right)+1} \\
&\leq \frac{\gamma\chi_2(\omega_1+2\omega_2+\omega_3)}{\left(\mathrm{Y}_S^2+\mathrm{Y}_{D_2}^2\right)\gamma(\omega_1+2\omega_2+\omega_3)+1}.
\end{aligned}
\tag{A22}
$$

From (A21) and (A22), we can obtain (22).
The proof is completed. $\square$

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
