# Peer review of "Ergodic Performance Analysis of Double Intelligent Reflecting Surfaces-Aided NOMA–UAV Systems with Hardware Impairment"

_drones, doi:10.3390/drones6120408_

Round 1

Reviewer 1 Report

In this manuscript, authors propose the design of intelligent reflecting surface-assisted Internet of Things by enabling non-orthogonal multiple access. The coverage of areas of interest is done by pair of UAV. The hypothesis is well mathematically backed and explained, authors team also provides proof of their calculations in appendixes. The hypothesis was also widely tested by simulations.

The manuscript is well organized, and its length is adequate.

The introduction provides sufficient background, and actual references appropriately cite it. Results are clearly presented, and conclusions are supported by the research.

Because the authors did not use a template, I could not comment on its compliance.

Author Response

 In this manuscript, authors propose the design of intelligent reflecting surface-assisted Internet of Things by enabling non-orthogonal multiple access. The coverage of areas of interest is done by pair of UAV. The hypothesis is well mathematically backed and explained, authors team also provides proof of their calculations in appendixes. The hypothesis was also widely tested by simulations. 

 The manuscript is well organized, and its length is adequate. 

 The introduction provides sufficient background, and actual references appropriately cite it. Results are clearly presented, and conclusions are supported by the research. 

 Comment 1: Because the authors did not use a template, I could not comment on its compliance. 

 Author Response: Thanks for your nice comment. We revised the paper following format provided by MDPI accordingly. 

Reviewer 2 Report

1. BS should send signals to UAV1 and UAV2, the communication channels should be different from the person on the ground. Also, why the signals can not directly to the person on the ground? it should be more explained. 

2. It is in one ideal case, different signals form UAVs should be arrived at different time, the time delay and interferences should be more considered in the paper. 

3. The main problem in the research should be focus on the UAVs, and the channels should be different from the  BS to Person.  4. Also, why use the UAVs in the paper, should be more explained in the paper

Author Response

 Comment 1: BS should send signals to UAV1 and UAV2, the communication channels should be different from the person on the ground. Also, why the signals can not directly to the person on the ground? it should be more explained. 

 Author Response: Thanks for your helpful suggestion. We redraw the system model. Since IRS is a meaningful device to reflect signals from the base station to ground users when users can not receive signals directly from the base station due to deep fading or high buildings. The channels representing different links are totally different. You can verify this in the following Matlab code. 

loop = 1e5; (the number of simulation trials)
N= 200; (reflecting elements of the IRS)

hSI1,n = sqrt(1/2) * (randn(N,loop)+1i*randn(N,loop)); (Rayleigh fading channel) 

hI1D1,n = sqrt(1/2) * (randn(N,loop)+1i*randn(N,loop)); 

hI1D2,n = sqrt(1/2) * (randn(N,loop)+1i*randn(N,loop)); 

hSI2,n = sqrt(1/2) * (randn(N,loop)+1i*randn(N,loop)); 

hI2D1,n = sqrt(1/2) * (randn(N,loop)+1i*randn(N,loop)); 

hI2D2,n = sqrt(1/2) * (randn(N,loop)+1i*randn(N,loop)); 

The Chanel S-IRS 1-D1 link:

A1^{1} = d_{SI1}^{ - \alpha /2}*d_{I1D1}^{ - alpha /2} *{\eta}*sum(hSI1,n*hI1D1,n,1);

The Chanel S-IRS 2-D1 link: 

A2^{1} = d_{SI2}^{ - \alpha /2}*d_{I2D1}^{ - \alpha /2}*\eta  *sum(hSI2,n*hI2D1,n,1);
The Chanel S-IRS 1-D2 link: 

B1^{1} = d_{SI1}^{ - \alpha /2}*d_{I1D2}^{ - \alpha /2}*\eta *sum(hSI1,n*hI1D2,n,1);

The Chanel S-IRS 2-D2 link: 

B2^{1} = d_{SI2}^{ - \alpha /2}*d_{I2D2}^{ - \alpha /2}*\eta *sum(hSI2,n*hI2D2,n,1);

 S-IRS 1-D1 link + S-IRS 2-D1 link: 

H1 = abs(A1^{1} + A2^{1})^2;

 S-IRS 1-D2 link + S-IRS 2-D2 link: 

H2 = abs(B1^{1} + B2^{1})^2;

 Comment 2: It is in one ideal case, different signals form UAVs should be arrived at different time, the time delay and interferences should be more considered in the paper. 

 Author Response: Thanks for your nice comment. We consider that time slot when the base station sends signals to group of users. Since the RIS is passive device, the IRS only reflects signals and it does not make interference to other devices. The delay is assumed to be short when we deal with short distances from the base station to users. We will study delay concern in future work. 

 Comment 3: The main problem in the research should be focus on the UAVs, and the channels should be different from the BS to Person. 

 Author Response: Thanks for your nice comment. We wanted to confirm that channels are created randomly and they are different completely. They are just in same kinds of distributions, either Rayleigh or Rician. 

 The distributions for Rayleigh fading channel:
loop = 1e5;
N= 200; (reflecting elements of the IRS)

hSI1,n = sqrt(1/2) * (randn(N,loop)+1i*randn(N,loop));

 The distributions for Rician fading channel: 

 KSI1 = 10 (dB);
Omega = 1; 

Sigma = sqrt(Omega./(2* (KSI1+1))); 

mu = sqrt(Omega .* (KSI1 ./ (KSI1 + 1) ) ); 

hSI1,n = random(’Rician’,mu,(Sigma),[N,1e6]); 

Comment 4: Also, why use the UAVs in the paper, should be more explained in the paper. 

 Author Response: We explained the role of UAV to extend the coverage of base station. We can deploy UAVs in smart cities’ applications. 

Reviewer 3 Report

By using non-orthogonal multiple access (NOMA) and unmanned aerial vehicle (UAV) methods, the paper designs Intelligent reflective surfaces (IRS) to assist the Internet of Things (IoT). For the deployment of low-cost iot systems, the dual IRS model is used as a reliable approach under the premise of controlling hardware loss. This scheme show that the proposed scheme can greatly improve achievable rates, obtain optimal performance at one of two devices and exhibit small performance gap.

1.The paper considers an IRS-NOMA-UAV system without a direct link, focusing on the performance analysis of two user groups and further determining the effects of hardware damage.

2.The paper gives the exact expressions of the achievable rate, deduces their simple approximations, and determines the influence parameters that determine the achievable rate.

Suggestion for modification:

1. The paper description expression can be further refined and improved. Important analytical methods and models need to be mentioned in the abstract. Examples include the channel models of Rayleigh and Rician.

2. Performance analysis of the IRS-NOMA-UAV system can be more intuitively demonstrated by comparing with other methods, such as IRS-OMA-UAV.

3. This paper uses Monte Carlo simulation to analyze the interrupt probability of the system. It can be considered to add simulation verification methods to analyze the system together.

4. When discuss the UAV tasks and optimization problems in introduction, the following literatures are of reference value, ‘Dynamic Reallocation Model of Multiple Unmanned Aerial Vehicle Tasks in Emergent Adjustment Scenarios’ and ‘A review on representative swarm intelligence algorithms for solving optimization problems: Applications and trends’.

5. The paper frame structure can be further optimized. When the simulation results are displayed, pay attention to the relationship between paragraph description and picture placement.

Suggestions: It is suggested to revise the paper carefully.

Author Response

 RESPONSE TO REVIEWER 3

 By using non-orthogonal multiple access (NOMA) and unmanned aerial vehicle (UAV) methods, the paper designs Intelligent reflective surfaces (IRS) to assist the Internet of Things (IoT). For the deployment of low-cost iot systems, the dual IRS model is used as a reliable approach under the premise of controlling hardware loss. This scheme show that the proposed scheme can greatly improve achievable rates, obtain optimal performance at one of two devices and exhibit small performance gap.

 The paper considers an IRS-NOMA-UAV system without a direct link, focusing on the performance analysis of two user groups and further determining the effects of hardware damage. 

 The paper gives the exact expressions of the achievable rate, deduces their simple approximations, and determines the influence parameters that determine the achievable rate. 

 Suggestion for modification: 

 Comment 1: The paper description expression can be further refined and improved. Important analytical methods and models need to be mentioned in the abstract. Examples include the channel models of Rayleigh and Rician. 

 Author Response: Thanks for your positive comment. We modified the way to present the expressions. Also, we updated detailed analytical methods and models in the abstract. 

 Comment 2: Performance analysis of the IRS-NOMA-UAV system can be more intuitively demonstrated by comparing with other methods, such as IRS-OMA-UAV.

 Author Response: Thanks for your nice comment. We made new numerical result to compare between IRS-NOMA-UAV and IRS-OMA-UAV. Please see the new figure 10.

 Comment 3: This paper uses Monte Carlo simulation to analyze the interrupt probability of the system. It can be considered to add simulation verification methods to analyze the system together.

 Author Response: Thanks for your nice comment. We added the flowchart to explain how we verify expressions via Monte-Carlo simulation.

 Comment 4: When discuss the UAV tasks and optimization problems in introduction, the following literatures are of reference value, ‘Dynamic Reallocation Model of Multiple Unmanned Aerial Vehicle Tasks in Emergent Adjustment Scenarios’ and ‘A review on representative swarm intelligence algorithms for solving optimization problems: Applications and trends’.

 Author Response: Thanks for your nice comment. We add these references. 

 Comment 5: The paper frame structure can be further optimized. When the simulation results are displayed, pay attention to the relationship between paragraph description and picture placement.

 Author Response: Thanks for your nice comment. We arranged the positions of numerical results and their explanations accordingly. 
